# Proteogenomic analysis of cancer aneuploidy and normal tissues reveals divergent modes of gene regulation across cellular pathways

Pan Cheng[1†], Xin Zhao[1†], Lizabeth Katsnelson[1‡], Elaine M Camacho-Hernandez[1‡], Angela Mermerian[1], Joseph C Mays[1], Scott M Lippman[2], Reyna Edith Rosales-Alvarez[3,4,5], Raquel Moya[1,6], Jasmine Shwetar[1], Dominic Grun[3,7], David Fenyo[1], Teresa Davoli[1*]

[1]Institute for Systems Genetics and Department of Biochemistry and Molecular Pharmacology, NYU School of Medicine, New York, United States; [2]Moores Cancer Center, University of California San Diego, La Jolla, United States; [3]Würzburg Institute of Systems Immunology, Max Planck Research Group at the Julius-Maximilians-Universität Würzburg, Würzburg, Germany; [4]International Max Planck Research School for Immunobiology, Epigenetics, and Metabolism, Freiburg, Germany; [5]Faculty of Biology, University of Freiburg, Freiburg, Germany; [6]Department of Pathology, NYU School of Medicine, New York, United States; [7]Helmholtz Institute for RNA-based Infection Research (HIRI), Helmholtz-Center for Infection Research, Würzburg, Germany

*For correspondence: teresa.davoli@nyulangone.org

[†]These authors contributed equally to this work
[‡]These authors also contributed equally to this work

**Abstract** How cells control gene expression is a fundamental question. The relative contribution of protein-level and RNA-level regulation to this process remains unclear. Here, we perform a proteogenomic analysis of tumors and untransformed cells containing somatic copy number alterations (SCNAs). By revealing how cells regulate RNA and protein abundances of genes with SCNAs, we provide insights into the rules of gene regulation. Protein complex genes have a strong protein-level regulation while non-complex genes have a strong RNA-level regulation. Notable exceptions are plasma membrane protein complex genes, which show a weak protein-level regulation and a stronger RNA-level regulation. Strikingly, we find a strong negative association between the degree of RNA-level and protein-level regulation across genes and cellular pathways. Moreover, genes participating in the same pathway show a similar degree of RNA- and protein-level regulation. Pathways including translation, splicing, RNA processing, and mitochondrial function show a stronger protein-level regulation while cell adhesion and migration pathways show a stronger RNA-level regulation. These results suggest that the evolution of gene regulation is shaped by functional constraints and that many cellular pathways tend to evolve one predominant mechanism of gene regulation at the protein level or at the RNA level.

## Editor's evaluation

The manuscript is of broad interest to researchers in the field of gene expression regulation and especially gene expression regulation in cancer cells. Gene expression can be regulated at several levels – in particular, the RNA and protein level. How each regulatory layer contributes to the final gene expression level is a central question in molecular biology. The authors tackle this fundamental question by asking how copy number variations at the level of DNA impact the other expression

layers of RNA and protein. They do so mainly in a huge cohort of cancer samples, but also show that their findings extend to untransformed cells, and they find that there is rarely compensatory regulation at the RNA and protein level together, but that depending on the gene, expression is either compensated at the RNA level or protein level. This is an extensive meta-analysis of a huge set of samples that will be of interest to a broad readership.

## Introduction

The expression level of each gene depends on the regulation of its transcript abundance (RNA-level regulation) and of its protein abundance (protein-level regulation) through synthesis, processing, and degradation of its transcript and protein. RNA-level and protein-level regulation is tightly controlled not only to adapt to changes in environmental conditions, but also as a mechanism to optimize energy consumption (*Franks et al., 2017*; *Wagner, 2005*). Certain genes are thought to have a predominant mechanism of regulation, either at the RNA level or at the protein level. For example, *HTERT*, encoding human telomerase, has a strong RNA-level regulation through transcriptional and splicing control (*Cong et al., 2002*; *Lazzerini-Denchi and Sfeir, 2016*). In contrast, *GCN4* (and its homolog *ATF4*) has a strong protein-level regulation through increased translation under endoplasmic reticulum (ER) stress (*Holcik and Sonenberg, 2005*). In addition, cell cycle genes such as *CDT1* and *CDC25A* are strongly regulated at the protein level through protein degradation (*Emanuele et al., 2011*). The relative contribution of RNA-level and protein-level regulation to control the expression level of human genes is currently incompletely understood (*Buccitelli and Selbach, 2020*).

To assess gene regulation, several studies have investigated the RNA and protein half-lives or the association between RNA and protein abundance across human genes in cells or tissues (*Gygi et al., 1999*; *Marguerat et al., 2012*; *Mathieson et al., 2018*; *McShane et al., 2016*; *Schwanhäusser et al., 2011*). Another way to investigate gene regulation is to measure how RNA and protein abundances change upon alterations in DNA copy number (somatic copy number alterations [SCNAs]) of a given gene that naturally occur in human cancers or that are experimentally engineered in cells. Previous proteomics analyses of aneuploid yeast and human cells have shown that, while most genes exhibit high correlation between DNA copy number and RNA abundance, a significant fraction of genes (20–30%) do not show protein abundance changes that are proportional to the DNA or RNA changes (*Gonçalves et al., 2017*; *Jovanovic et al., 2015*; *McShane et al., 2016*; *Stingele et al., 2012*; *Torres et al., 2007*). In other words, the change of protein abundance is less than what it is expected based on its DNA change; this phenomenon is referred to as gene compensation. In particular, genes whose products are participating in protein complexes (protein complex genes) show a stronger compensation than genes that are not part of complexes (non-complex genes). Importantly, a recent study showed that the genes showing compensation at the protein level in aneuploid cells also have a high degree of protein-level regulation in normal diploid cells (*McShane et al., 2016*). In other words, this study found that protein compensation in aneuploid cells is associated with protein regulation (e.g., degradation patterns) in normal cells. This supports the fact that studying how protein levels are affected by SCNAs in aneuploid cancer cells can inform us on how genes are regulated in normal non-aneuploid cells (*Taggart et al., 2020*).

Although such studies have advanced our understanding of how cells regulate RNA and protein abundances of genes that contain SCNAs, several outstanding questions remain. For example, is gene compensation after SCNAs similar across tissue types? How do biological pathways, cellular localization and gene evolution influence the mechanism of gene regulation? Can we use SCNA analysis to investigate not only protein-level regulation but also RNA-level regulation and the relationship between the two? Here we perform a proteogenomic analysis (analysis of DNA, RNA and protein levels) across primary tumor samples and cancer cell lines from different tumor types, a panel of isogenic non-tumorigenic human colon epithelial cells (hCECs) and normal tissues. We find tissue specificity in the RNA-level and protein-level compensation of genes affected by SCNAs. Importantly, we then utilize the DNA–RNA and RNA–protein correlations to infer the degree of regulation at the RNA and protein levels, respectively. In fact, as RNA–protein correlation informs us on the protein-level regulation, DNA–RNA correlation can inform us on the RNA-level regulation assuming DNA alterations are equally variable (as discussed below, see *Figure 2—figure supplement 1*). Protein complex genes have a stronger protein-level regulation, while non-complex genes show stronger RNA-level

regulation. Strikingly, we found an inverse relationship between the degree of RNA-level regulation and the degree of protein-level regulation across genes and cellular pathways. This suggests that cellular function impacts gene regulation and, for several pathways, tends to favor either RNA- or protein-level regulation. Finally, genes involved in RNA processing, translation, and mitochondrial regulation are upregulated in highly aneuploid primary tumor samples (compared to low aneuploid tumors), especially at the protein level.

## Results

### Analysis of gene compensation at RNA and protein levels across tumor types

Gene compensation is a process by which cells modulate gene expression to buffer against changes in DNA copy number. In order to assess the degree of compensation at the RNA or protein level after DNA gains or losses (*Figure 1A*), we used the Clinical Proteomic Tumor Analysis Consortium (CPTAC) dataset, a compendium of thousands of tumor samples analyzed for their genomic, transcriptomic, and proteomic features (*Ang et al., 2019*). Here, we analyzed CPTAC data comprising about 700 tumor samples derived from seven tumor types: colon adenocarcinoma (COAD), breast cancer (BRCA), ovarian cancer (OV), clear cell renal cell carcinoma (ccRCC), uterine corpus endometrial carcinoma (UCEC), HPV-negative head and neck squamous cell carcinoma (HNSC), and lung adenocarcinoma (LUAD). The dataset contains information at the DNA level by whole-genome sequencing (WGS) or whole-exome sequencing (WES), RNA level by RNAseq and protein level by TMT mass spectrometry for 7–12K genes (*Supplementary file 1A*).

For each gene of each cancer type, we defined the samples that did not have DNA copy number changes (log2 copy number ratio, defined as the log2 of the ratio between the copy number of the gene and the average copy number of the rest of the genome, between –0.2 and 0.2) as the neutral group. We considered the median of the DNA, RNA, and protein amount of this neutral group as the *neutral* DNA, RNA, or protein level. Then, we calculated the log2 fold change (log2FC) of the DNA, RNA, and protein amount for each gene in each tumor sample relative to the corresponding *neutral* levels. We next determined the distributions of DNA, RNA, and protein log2FC of all genes from the seven tumor types, based on five groups of DNA change (i.e., deep loss [DNA log2FC < –0.65]; loss [–0.65 < DNA log2FC < –0.2]; neutral [–0.2 < DNA log2FC < 0.2]; gain [0.2 < DNA log2FC < 0.65]; and high gain [DNA log2FC > 0.65]). Within each of these five groups, we also split genes into protein complex genes and non-complex genes based on the CORUM database (*Ruepp et al., 2008*; see below, *Figure 1B*). In order to quantify the degree of RNA- or protein-level compensation, we calculated a compensation score (CS) for each gene in each sample, determined as the difference between the RNA or protein log2FC with the DNA log2FC (see 'Methods'). To assess whether there was significant compensation in each group of DNA change (i.e., CS was significantly larger than zero), we implemented a bootstrapping method by randomly sampling the CS of genes within each group.

Work done in model organisms and isogenic human cells suggests that RNA levels change proportionally to the DNA levels in aneuploid cells (i.e., there is no RNA-level compensation) while the protein levels do not (i.e., there is protein-level compensation) (*Oromendia et al., 2012*; *Stingele et al., 2012*; *Torres et al., 2010*). In our pan-cancer analysis (*Figure 1B*), this was generally the case, with a significant protein-level compensation (false discovery rate [FDR] < 0.001) and no significant RNA-level compensation except in the high gain group (see also below; FDR < 0.001; *Supplementary file 1B*). Protein-level compensation was significant for both gains and losses, although it was stronger in the high gain group versus the deep loss group (FDR < 0.001).

Despite these overall trends in the pan-cancer analysis, we observed tissue specific patterns when we conducted the same analysis for each tumor type (*Figure 1C and D*, *Supplementary file 1C*). While protein-level compensation was widespread, LUAD did not show protein-level compensation (FDR = 1, *Supplementary file 1C*) and BRCA showed reduced protein-level compensation compared to other cancer types, especially for DNA losses (*Supplementary file 1C*). Surprisingly, we found significant RNA-level compensation in certain tumor types. COAD showed general RNA-level compensation both for gains and losses (FDR < 0.001, *Supplementary file 1C*; degree of compensation was lower for deep loss than other SCNA groups). In addition, ccRCC showed RNA-level compensation for deep losses (FDR < 0.001), and BRCA exhibited RNA-level compensation for high gains (FDR < 0.001).

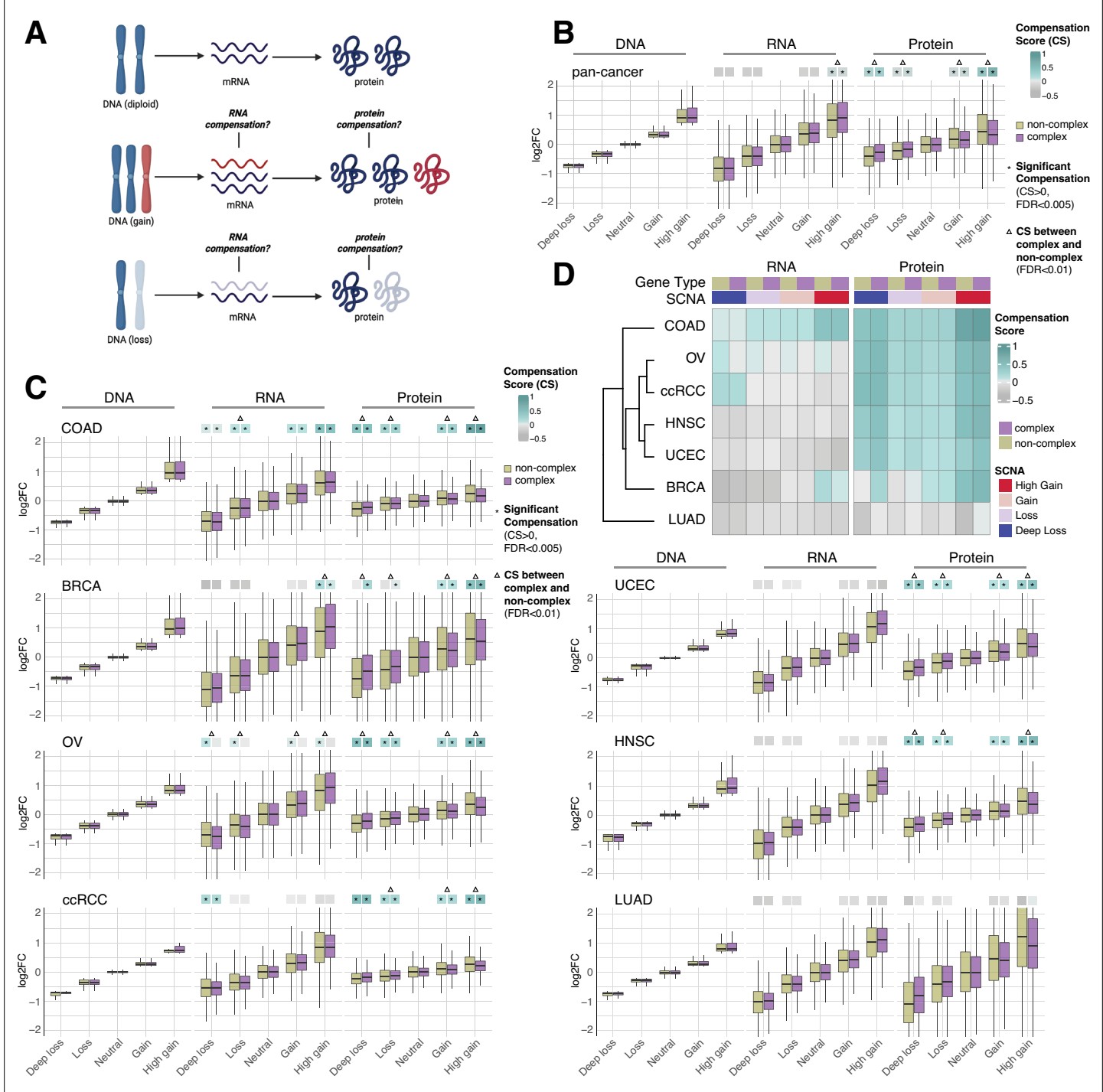

**Figure 1.** RNA-level and protein-level gene compensation after somatic copy number alteration (SCNA) across tumor types. (**A**) Schematic of RNA-level and protein-level gene compensation as a result of DNA gains (red) or losses (light blue). RNA and protein abundance change proportionally to the DNA change when gene compensation is absent. (**B**) Box plots showing the Clinical Proteomic Tumor Analysis Consortium (CPTAC) pan-cancer profile of DNA, RNA, and protein log2 fold change (log2FC) in five groups based on the copy number change: deep loss, loss, neutral, gain, and high gain. The genes of each group were separated in protein complex genes (purple) and non-complex genes (yellow). The median of the compensation score (CS) in each condition, which represents the degree of gene compensation, is shown at the top of the box plot (cyan/gray squares). CS is positive when compensation happens (cyan) and is proportional to the degree of compensation. To test whether CS were significantly positive, we used a bootstrapping test and p-values were corrected for false discovery rate (FDR). An asterisk in the square indicates significant CS (CS > 0 and FDR < 0.005). A triangle above the squares indicates that the CS of complex and non-complex genes is significantly different by bootstrapping test (FDR < 0.005). (**C**) Box plots showing the profiles of DNA, RNA, and protein log2FC of the indicated cancer types grouped in five groups as in (**B**). The median

*Figure 1 continued on next page*

*Figure 1 continued*

CS is shown at the top of the box plots (cyan/gray squares). An asterisk in the square represents significant compensation (CS > 0 and FDR < 0.005). A triangle above the squares indicates that the CS of complex and non-complex genes is significantly different by bootstrapping test (FDR < 0.005). (**D**) Heatmap showing the RNA-level and protein-level CS of different cancers. Cancers were clustered by Euclidean distance and hierarchical clustering. For all box plots, box sizes represent the interquartile range (IQR), whiskers expand to± 1.5*IQR of the box limits, and outliers beyond the whisker limits are not shown.

The online version of this article includes the following figure supplement(s) for figure 1:

**Figure supplement 1.** Gene compensation is not biased by genes encoding ribosome subunits, technical limitations of proteome detection, or genome doubling.

**Figure supplement 2.** Gene compensation at the RNA and protein levels in cancer cell lines and isogenic human colon epithelial cell (hCEC).

Interestingly, OV showed RNA-level compensation for non-complex genes but not for complex genes in all SCNA groups (*Figure 1C and D*, *Supplementary file 1C*, see below).

Next, for each SCNA group (loss, deep loss, gain, high gain), we compared the genes belonging to protein complexes ('CORUM,' composed of 3449 protein complex genes) with those that do not ('NoCORUM,' non-complex genes, i.e., remaining genes) (*Ruepp et al., 2008*; *Figure 1B–D*). In general, we found that protein complex genes had a stronger protein-level compensation compared to non-complex genes (*Figure 1B*, FDR < 0.001, *Supplementary file 1D*), consistent with previous studies examining the effect of chromosome gains (*Oromendia et al., 2012*; *Stingele et al., 2012*; *Torres et al., 2010*). Importantly, this was true not only for gains, but also for losses (*Figure 1B*, FDR < 0.001, *Supplementary file 1D*) and across tumor types (*Figure 1C*, FDR < 0.001, *Supplementary file 1E*). Interestingly, at the RNA level, the opposite was true: protein complex genes showed less RNA-level compensation compared to non-complex genes for high DNA gain, the only group showing significant compensation in the pan-cancer analysis (*Figure 1B*, FDR < 0.001, *Supplementary file 1D*). As mentioned above, for the individual tumor types, only certain cancers showed significant RNA-level compensation; in the majority of those cases, complex genes showed less RNA-level compensation compared to non-complex genes (high gain group of BRCA and all groups of OV, *Figure 1C*, FDR < 0.001, *Supplementary file 1E*). For example, OV showed significant RNA-level compensation for non-complex genes but not for complex genes across all DNA groups (*Figure 1C*, FDR < 0.001, *Supplementary file 1C*). In other words, protein complex genes showed changes at the RNA level that were more similar in amplitude to the changes observed at the DNA level than non-complex genes, implying a lower level of regulation at the RNA level (see next section).

Since ribosomal subunits make up a significant fraction of protein complexes and are synthesized in large amounts in cells, we investigated whether our results related to the regulation of protein complex genes are dependent on the presence of ribosomal genes among protein complex genes. We split complexes genes into ribosomal genes and non-ribosomal complex genes. Both ribosomal and non-ribosomal complex genes showed significant compensation at the protein level for both gains and losses, indicating that our results are not dependent on ribosomal genes (*Figure 1—figure supplement 1B*, *Supplementary file 1F*). Notably, the protein-level compensation of ribosomal genes was so strong that the median protein log2FC remained almost unchanged for high gains and deep losses; this was not the case for the RNA level (*Figure 1—figure supplement 1B*, *Supplementary file 1F*).

Another potential factor that may hinder the accurate inference of gene compensation is from the technical limitation of tandem mass tag (TMT) mass spectrometry. TMT-based proteome quantifications, widely adopted in CPTAC database, suffer from the issue of ratio compression and may underestimate the actual change (*Savitski et al., 2013*). Nevertheless, previous studies using other techniques such as stable isotope labeling with amino acids in cell culture (SILAC) or MS3-based proteomics found widespread protein compensation for complex genes after DNA gain in yeast and pairs of isogenic diploid and aneuploid cell lines, consistent with our observations (*Dephoure et al., 2014*; *Hwang et al., 2021*; *Stingele et al., 2012*). To further exclude the impact of ratio compression on our results, we performed the analysis shown in *Figure 1* on The Cancer Genome Atlas (TCGA; *Weinstein et al., 2013*) COAD samples for which label-free proteomics data is available (*Zhang et al., 2014*). Consistent with TMT-based proteomics, significant compensation at the protein level was found, which is higher for complex genes than non-complex genes (*Figure 1—figure supplement 1C*, *Supplementary file 1G*). As we observed before, for COAD (*Figure 1C*), RNA-level compensation was shown in all groups of DNA change and was stronger for non-complex genes (deep loss and

high gain, FDR < 0.005, *Figure 1—figure supplement 1C*, *Supplementary file 1G*). These additional observations indicate that the limitations imposed by the TMT quantification do not affect the results of our analyses.

Another possible confounder is genome doubling, which is common in cancer and may affect the calculation of relative changes of DNA, RNA, or protein. However, most CPTAC databases lack genome doubling information. To exclude the interference of genome doubling, we analyzed the proteomics data for TCGA samples (*Mertins et al., 2016*; the *Zhang et al., 2014*; *Zhang et al., 2016*), for which this information is available. Samples inferred by ABSOLUTE (*Carter et al., 2012*) to have undergone genome doubling, were removed from the analysis. Consistent with *Figure 1C*, widespread protein compensation was observed for complex genes, and RNA compensation was observed for non-complex genes (*Figure 1—figure supplement 1D*, *Supplementary file 1H*). Therefore, these data indicate that the presence of genome doubling in a fraction of the samples does not affect the results of our analyses.

To validate the findings observed from primary tumors, we performed the same analysis on cancer cell lines using the Cancer Cell Line Encyclopedia (CCLE) (*Barretina et al., 2012*; *Nusinow et al., 2020*), which showed general protein-level compensation and negligible RNA-level compensation except in the deep loss group at the pan-cancer level (*Figure 1—figure supplement 2A*, FDR < 0.001, *Supplementary file 1I*). Protein complex genes had stronger protein-level compensation than non-complex genes similar to primary tumors (*Figure 1—figure supplement 2A*, FDR < 0.001, *Supplementary file 1J*). To further test this observation, we generated a panel of isogenic immortalized non-transformed human colon epithelial cells (hCEC) with different aneuploidy patterns (*Figure 1—figure supplement 2B and C*, *Supplementary file 1K*). We treated hTERT-immortalized TP53-KO (non-tumorigenic) hCEC (*Ly et al., 2011*; *Sack et al., 2018*) with reversine, an MPS1 inhibitor that inhibits correct chromosome attachment and spindle checkpoint, to induce random chromosome missegregation and subsequent aneuploidy (*Santaguida et al., 2015*). Clones derived from single cells contained different patterns of aneuploidy, characterized by WGS (*Figure 1—figure supplement 2C*, *Supplementary file 1K*). We analyzed their transcriptome and proteome using RNA-sequencing and TMT mass spectrometry, respectively (see 'Methods'). Interestingly, in addition to the widespread protein-level compensation, hCEC also showed RNA-level compensation as COAD did (*Figure 1—figure supplement 2D*, FDR < 0.001, *Supplementary file 1L*). Similar to COAD primary tumors, in hCEC complex genes had stronger protein-level compensation for the DNA gain and deep loss groups but weaker RNA-level compensation for the DNA loss group (*Figure 1—figure supplement 2D*, FDR = 0.013, *Supplementary file 1M*). The accuracy of RNA log2FC calculated from RNAseq was validated by qPCR for representative genes (*Supplementary file 1N*).

Overall, these data suggest that while protein-level compensation is widespread and RNA-level compensation is virtually absent in our pan-cancer analysis, there is significant tissue specificity especially in the presence and degree of RNA-level compensation. Indeed, some tissue types (such as lung cancer) show low levels of compensation both at the RNA and protein level, while others (such as colon, breast, ovarian, and renal cancers) show unexpectedly high compensation at the RNA level. Furthermore, protein complex genes generally showed stronger protein-level compensation and weaker RNA-level compensation compared to non-complex genes.

## Protein complex genes have a higher protein-level regulation and a lower RNA-level regulation than non-complex genes

Our previous analysis (*Figure 1B–D*) suggested that protein complex genes have stronger protein-level compensation (and thus regulation) and weaker RNA-level compensation (and thus regulation) compared to non-complex genes. To better understand this phenomenon, we decided to systematically study the correlation between DNA and RNA levels and between RNA and protein levels for each gene across samples, to infer the degree of gene regulation at the RNA and protein levels, respectively (*Figure 2A*). In other words, if the correlation between RNA and protein is very high, we can infer that the protein abundance is mainly determined by the RNA amount with minimal protein-level regulation. On the other hand, if the correlation between RNA and protein is low, we assume a strong level of protein-level regulation. A similar logic can be used to infer the RNA-level regulation based on the DNA–RNA correlation. To do this, we calculated Spearman's correlation coefficients (rho) between DNA and RNA levels and between RNA and protein levels for each gene across tumor samples

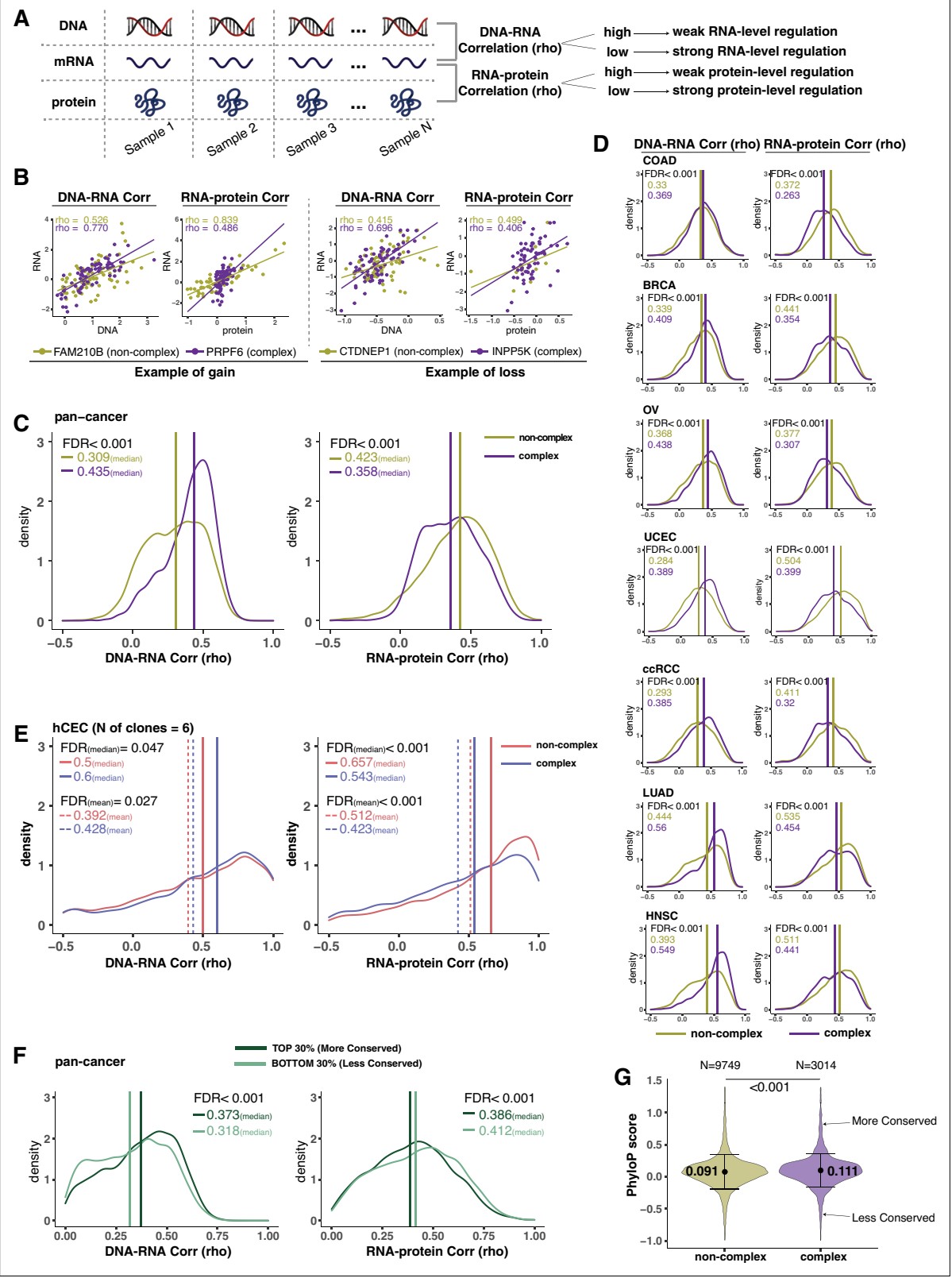

**Figure 2.** Protein complex genes have a stronger protein-level regulation but a weaker RNA-level regulation than non-complex genes. (**A**) Schematic representing the strategy to infer the degree of RNA- or protein-level regulation by Spearman's correlation analysis between DNA and RNA (DNA–RNA) or between RNA and protein (RNA–protein). A high (versus low) correlation indicates a weak (versus strong) regulation. (**B**) DNA–RNA and RNA–protein correlations for representative complex and non-complex genes frequently gained (FAM210B and PRPF6) or lost (CTDNEP1 and INPP5K) in colon

*Figure 2 continued on next page*

*Figure 2 continued*

adenocarcinoma (COAD). Dots represent different samples; solid lines indicate the linear regression line between DNA–RNA and RNA–protein; Spearman's correlation is shown for each gene. (**C**) Density distribution of DNA–RNA and RNA–protein correlations for pan-cancer analysis (protein complex genes in purple and non-complex genes in golden yellow). Vertical lines and numbers in the top left represent the median correlation of protein complex genes or non-complex genes. Difference of the median correlation coefficients between protein complex genes and non-complex genes was evaluated by bootstrapping, and p-values were adjusted for false discovery rate (FDR). (**D**) Density distribution of DNA–RNA and RNA–protein correlations for individual Clinical Proteomic Tumor Analysis Consortium (CPTAC) cancer types as in (**C**). (**E**) Density distribution of DNA–RNA and RNA–protein correlations for human colon epithelial cell (hCEC) cell lines (protein complex genes in blue and non-complex genes in red). Blue/red vertical solid (or dashed) lines and numbers in the top left represent the median (or mean) correlation of protein complex or non-complex genes. Difference of the median correlation coefficients between protein complex genes and non-complex genes was evaluated by bootstrapping, and p-values were adjusted for FDR. (**F**) Density distribution of DNA–RNA and RNA–protein correlations for evolutionally more conserved genes (dark green; genes in the top 30% of phyloP scores) and less conserved genes (light green; genes in the bottom 30% of phyloP scores). Dark green or light green vertical lines and numbers in the top right represent the median of the more conserved or less conserved genes, respectively. Difference of the median correlation coefficients between more and less conserved genes was evaluated by bootstrapping, and p-values were adjusted for FDR. (**G**) The phyloP score difference between protein complex genes and non-complex genes. Difference between protein complex genes and non-complex genes was evaluated by bootstrapping, and p-values were adjusted for FDR. The error bars represent standard deviation.

The online version of this article includes the following figure supplement(s) for figure 2:

**Figure supplement 1.** DNA–RNA and RNA–protein correlations across complex and non-complex genes among cell lines and normal tissues.

(*Figure 2B–D*). Since the correlation coefficient depends on the extent of variation, we excluded the genes that show very little or no changes at the DNA level across samples (–0.02 < log2 copy number ratio < 0.02 in more than 70% of the samples; see 'Methods'). We calculated the pan-cancer distributions of the correlation coefficients for complex and non-complex genes using the mean correlation coefficient value across the seven tumor types (*Figure 2C*, *Supplementary file 2A*). As expected, we found that the median of the RNA–protein correlations was significantly lower for protein complex genes than for non-complex genes (FDR < 0.001), indicating that protein complex genes tend to have stronger protein-level regulation compared to non-complex genes (*Stingele et al., 2012*). Strikingly, the opposite was true for RNA-level regulation, where the median of the DNA–RNA correlations was significantly higher for protein complex genes than for non-complex genes (FDR < 0.001). This was in agreement with the observations described above regarding compensation in complex and non-complex genes (*Figure 1B–D*). In addition, it was not due to the difference in the RNA abundance (*Figure 2—figure supplement 1A*; analysis repeated with the exclusion of genes of low RNA abundance) or difference in the variance of DNA alterations between protein complex and non-complex genes (*Figure 2—figure supplement 1B*; ANOVA across DNA values), or to the fact that ribosomal complex genes make up the majority of complex genes (*Figure 2—figure supplement 1C*; analysis repeated with the exclusion of ribosomal genes). This result indicates that protein complex genes are likely to have weaker RNA-level regulation compared to non-complex genes. We also note that in terms of absolute correlation values (Spearman's correlation), protein complex genes have lower RNA–protein correlations compared to DNA–RNA correlation values (median DNA–RNA correlation: 0.44; median RNA–protein correlation: 0.36 for protein complex genes, pan-cancer analysis), while it is the opposite for non-complex genes (median DNA–RNA correlation: 0.31; median RNA–RNA–protein correlation: 0.42 for non-complex genes, pan-cancer analysis, *Figure 2C*).

We next extended these analyses to the seven tumor types individually and observed that this result found in the pan-cancer analysis was recapitulated across all of them (FDR < 0.001, *Figure 2D*, *Supplementary file 2B–H*). Furthermore, this finding was confirmed using other proteogenomic datasets of cancer cell lines such as CCLE and NCI-60 (*Alley et al., 1988*; *Figure 2—figure supplement 1D*). For the DNA–RNA correlation analysis, we also used the TCGA dataset containing additional tumor types with DNA and RNA information and confirmed that DNA–RNA correlations were significantly higher for protein complex genes than for non-complex genes (*Supplementary file 2I*).

Next, we wanted to test whether these results obtained from primary tumors or cancer cell lines were recapitulated in non-tumor-derived cell lines and normal tissues. Using our panel of isogenic untransformed hCEC with different aneuploid patterns (*Supplementary file 1H*), we confirmed that the DNA–RNA correlation was significantly higher and the RNA–protein correlation was significantly lower among protein complex genes than those among non-complex (*Figure 2E*). Finally, we interrogated a database of normal tissues that includes RNA and protein levels for the RNA–protein correlations (DNA–RNA correlations were absent as SCNAs are generally not present in normal tissues; *Wang*

*et al., 2019*). Similarly, even in the normal tissues, we confirmed the lower RNA–protein correlation for complex genes compared to non-complex genes (*Figure 2—figure supplement 1E*). Altogether, these data indicate that protein complex genes have a stronger protein-level regulation and a weaker RNA-level regulation compared to non-complex genes. These results also indicate that our findings in tumors or tumor cell lines are recapitulated in untransformed isogenic aneuploid cells as well as normal tissues.

In addition to participation in protein complexes, we investigated other parameters, including biophysical properties and evolutionary conservation, for their association with gene regulation (DNA–RNA or RNA–protein correlation) (*Schukken and Sheltzer, 2022*). Some of these properties, including protein polyampholyte score, protein polarity, and protein aggregation score, had no significant association with the type of gene regulation (*Supplementary file 2J*). The non-exponential degradation score, a score representing the likelihood that a protein is degraded in a non-exponential way (*McShane et al., 2016*), was predictive of strong regulation at the protein level, consistent with previous findings (*McShane et al., 2016*; *Figure 2—figure supplement 1F*). Interestingly, the evolutionary conservation score (phyloP score) (*Hubisz et al., 2011*) was associated with the RNA-level and protein-level regulation. When we compared genes with high versus low conservation, we found that more conserved genes tended to have lower RNA–protein correlation and higher DNA–RNA correlation compared to less conserved genes, and thus stronger protein-level regulation and weaker RNA-level regulation (*Figure 2F and G*; FDR < 0.001). The conservation score was also significantly higher for protein complex genes than non-complex genes (*Figure 2G*, mean: 0.11 vs. 0.09, FDR < 0.001; variance: 0.31 vs. 0.34, p=0.004). Altogether, these data indicate that protein complex genes are more evolutionarily conserved, and that genes that are highly conserved tend to have a strong regulation at the protein level and vice versa (see 'Discussion').

## Negative association between RNA-level regulation and protein-level regulation across cellular pathways

We next performed a systematic analysis to understand whether and how gene function (i.e., belonging to a certain cellular pathway) and subcellular location influence the extent of RNA-level and protein-level regulation. This would inform us on whether genes may have evolved a preferred mechanism of gene regulation depending on the biological function or cellular distribution of the encoded protein. We started by examining the relationship between the DNA–RNA and the RNA–protein correlations across genes from different CPTAC tumors as a pan-cancer analysis. Interestingly, we found a significant negative association between these two parameters (slope = −0.33, rho = −0.78, p=7.9E-07 see 'Methods', *Figure 3A*). In other words, genes showing a high DNA–RNA correlation tend to have a low RNA–protein correlation and vice versa. Based on this finding, we next asked whether the genes residing at the two ends of the distribution (high DNA–RNA correlation and low RNA–protein correlation, or low DNA–RNA correlation and high RNA–protein correlation) show enrichment in specific biological function. To this end, we defined two main groups of genes using the DNA–RNA and RNA–protein correlations (pan-cancer analysis): Group 1, composed of genes with a high DNA–RNA correlation (top 35%, rho > 0.43) and a low RNA–protein correlation (bottom 35%, rho < 0.31), and Group 2 of genes with a low DNA–RNA correlation (bottom 35%, rho < 0.24) and a high RNA–protein correlation (top 35%, rho > 0.50) (*Supplementary file 3A*). Gene Ontology (GO) enrichment analysis showed that Group 1 was strongly enriched in mitochondrial pathways (*Supplementary file 3B-I*, *Figure 3—figure supplement 1A*; e.g., mitochondrial translation in pan-cancer; FDR = 4.5E-43), protein translation (*Figure 3—figure supplement 1A*, *Supplementary file 3B-I*; e.g., ribosome biogenesis; FDR = 3.3E-23 and cytoplasmic translation in pan-cancer; FDR = 1.0E-07), and RNA processing (*Supplementary file 3B-I*, *Figure 3—figure supplement 1A*; e.g., RNA splicing; FDR = 4.6E-29 and non-coding-RNA metabolism; FDR = 6.0E-23). On the other hand, Group 2 was enriched in cell structure (*Figure 3—figure supplement 1A*, *Supplementary file 3B-I*; e.g., actin filament organization; FDR = 9.4E-10), cell adhesion (*Figure 3—figure supplement 1A*, *Supplementary file 3B-I*; e.g., substrate adhesion, FDR = 2.1E-17 and matrix adhesion, FDR = 2.2E-11), and cell migration (FDR = 0.044). Analysis in individual tumor types was overall similar to the pan-cancer analysis results for the pathways enriched in Groups 1 and 2 (*Figure 3—figure supplement 1A*, *Supplementary file 3C-I*). Importantly, the result of the pathway enrichment analysis in the two groups of genes (Groups 1 and 2) was validated using the CCLE dataset (*Figure 3—figure supplement 1B*).

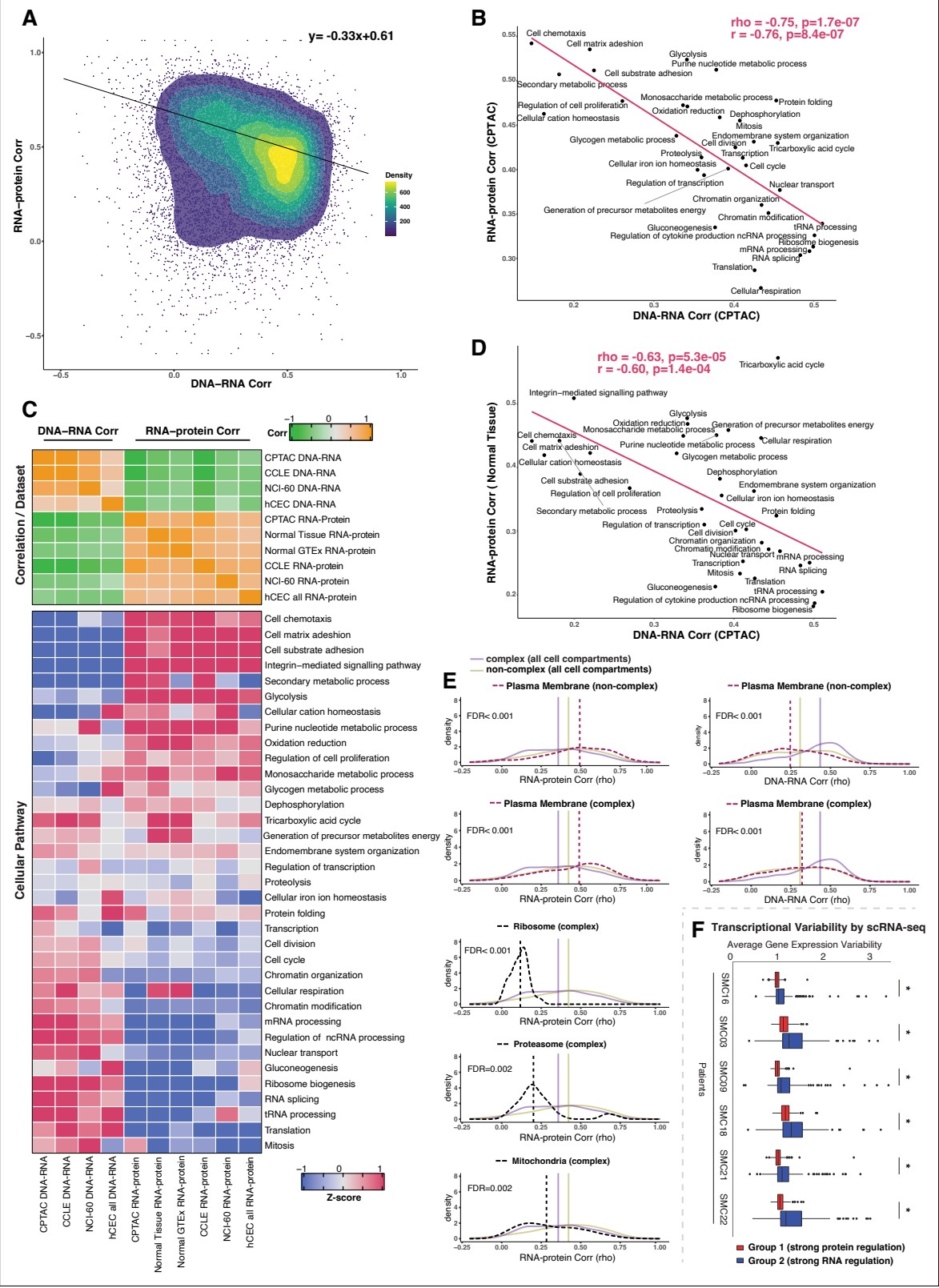

**Figure 3.** Negative association between RNA- and protein-level regulation across genes and pathways. (**A**) Dot plot showing the association between DNA–RNA and RNA–protein correlation, where each point is a gene. Density distribution is shown and a density distribution-dependent slope was calculated to estimate the association between the DNA–RNA and RNA–protein correlations. (**B**) A pathway-level analysis for the association between DNA–RNA and RNA–protein correlations (pan-cancer analysis, Clinical Proteomic Tumor Analysis Consortium [CPTAC]). The DNA–RNA

*Figure 3 continued on next page*

*Figure 3 continued*

(x-axis) and RNA–protein (y-axis) correlation for each cellular pathway was calculated using the median rho value across all genes belonging to the pathway (pathway database: msigdbr, v7.4.1, category = C5). Spearman's (rho) and Pearson's (r) correlation coefficients are shown at the top right of the panel. (**C**) Top panel: a heatmap showing Spearman's correlations among different proteogenomic datasets. For each dataset, we first calculated the DNA–RNA and RNA–protein rho values for each gene, and then we calculated the Spearman's correlation between these rho values (DNA–RNA rho or RNA–protein rho) across datasets. Bottom panel: a heatmap showing the pathway-level DNA–RNA and RNA–protein correlation score among different datasets. The pathway-level score was calculated by the median value across all genes in the same pathway and then Z-score transformed (pathway database: msigdbr, v7.4.1, category = C5). (**D**) A pathway-level analysis for the DNA–RNA (CPTAC) and RNA–protein (normal tissues, *Wang et al., 2019*) correlations. The DNA–RNA (x-axis) and RNA–protein (y-axis) correlation for each cellular pathway was calculated using the median rho value across all genes belonging to the pathway (pathway database: msigdbr, v7.4.1, category = C5). Spearman's (rho) and Pearson's (r) correlation coefficients are shown at the top right of the panel. (**E**) Density distribution of DNA–RNA correlations for genes belonging plasma membrane and RNA–protein correlations for genes belonging to ribosome, proteasome, mitochondria, and plasma membrane. The dashed line represents the specific cell compartment as indicated, the transparent purple or golden yellow line represents the median of complex (all cell compartments) or non-complex genes (all cell compartments). Significance between the genes in the specific cell compartment and complex or non-complex genes of all cell compartments was evaluated based on bootstrapping test and adjusted for false discovery rate (FDR). For example, the FDR in the top left panel was evaluated based on the difference between plasma membrane (non-complex) and complex genes from all cell compartments. (**F**) Gene-wise variability levels of scRNAseq data from Korean colorectal cancer patients (*Lee et al., 2020*) estimated by VarID (*Grün, 2020*). Genes were grouped according to their preferential regulation at the protein level (Group 1) or RNA level (Group 2). The averages of corrected variance estimates per gene are shown ('Methods'). Box sizes represent the interquartile range (IQR), whiskers expand to± 1.5*IQR of the box limits, and outliers beyond the whisker limits are also shown. p-adjusted value: *<0.001 (one-sided Wilcoxon test with Bonferroni correction).

The online version of this article includes the following figure supplement(s) for figure 3:

**Figure supplement 1.** Enriched gene sets among genes with different regulation in primary tumors and cell lines.

**Figure supplement 2.** DNA–RNA and RNA–protein correlations among genes localized in different cellular compartments.

Since the genes showing similar DNA–RNA and RNA–protein correlations were enriched for specific cellular pathways, we calculated the median value for these correlations among the genes in each pathway, thus obtaining pathway-level values for the DNA–RNA and RNA–protein correlations. For this analysis, we considered the cellular pathways used in a previous study (*Schwanhäusser et al., 2011*) their genes were identified by the msigdb Gene Set Enrichment Analysis (GSEA) database (v7.4). Altogether, the genes in these pathways accounted for 84% of all genes. Strikingly, we found a strong negative correlation between the pathway-level DNA–RNA and RNA–protein correlations (rho = −0.75, p = 1.7E-07; *Figure 3B*, *Figure 3—figure supplement 1C*), corroborating and extending the finding described above at the individual gene level. In agreement with our previous enrichment analysis (*Figure 3—figure supplement 1A*), RNA processing and translation pathways showed a preference for high DNA–RNA and low RNA–protein correlations while cell adhesion and matrix-related pathways tended to have high RNA–protein and low DNA–RNA correlations (*Figure 3B and C*). While this analysis was performed at the pan-cancer level, a similar result was obtained when we performed the analysis within each individual tumor type (*Supplementary file 3K*, significant negative correlation between the pathway-level DNA–RNA and RNA–protein correlations among all tumor types), indicating that it reflects a general property of gen regulation independent of tissue type. Importantly, these results were confirmed based on the CCLE and NCI-60 datasets and based on our isogenic hCEC data (*Figure 3C*, *Supplementary file 3J*). Furthermore, a significant negative association among pathways was maintained when we calculated it considering only complex genes (rho = −0.58, p = 0.00037 ) or non-complex genes (rho = −0.54, p = 0.00078) using the CPTAC dataset (as in *Figure 3B*), suggesting that it is not simply due to the different percentage of genes in protein complexes in each pathway (Supplementary File 3K, column 'Percentage of genes in protein complexes'). Finally, we also confirmed the negative correlation was not an artifact due to different variance in the DNA values across genes, that is, by the fact that the genes in certain pathways were more likely to be gained or lost than the genes in other pathways (*Supplementary file 3K*). In addition, we also observed that RNA half-life was positively associated with the RNA–protein correlation (rho = 0.51, p=0.001) and was negatively associated with the DNA–RNA correlation (rho = −0.52, p=0.002), while no such association was found for protein half-life (*Supplementary file 3J*, see 'Discussion').

A recent study showed that protein regulation in aneuploid cells is associated with protein regulation in normal cells (*McShane et al., 2016*), indicating that studying how protein levels are affected by SCNAs in aneuploid cancer cells can inform us on how genes are regulated in normal non-aneuploid cells. To test whether our main conclusion on gene regulation obtained from analyzing genes containing

SCNAs was recapitulated also in normal tissues, we utilized two independent proteogenomic datasets from normal tissues (*GTEx Consortium, 2013*; *Jiang et al., 2020*; *Wang et al., 2019*). Once again we found a significant negative association between the RNA–protein correlations calculated from normal tissues and the DNA–RNA correlations from CPTAC tumor tissues (rho = −0.63, p=5.3E-05; *Figure 3C and D*), indicating the inverse relationship between the RNA-level and protein-level regulation across cellular pathways is also present in normal tissues. Altogether, these results suggest that RNA-level and protein-level regulation tends to be negatively associated across genes and pathways. In other words, genes (and pathways) that have strong regulation at the RNA level generally do not have a strong regulation at the protein level and vice versa, perhaps due to an evolutionary selective pressure to favor one type of regulation over the other, depending on gene function (see 'Discussion').

Since specific biological pathways were predictive of whether a gene was more regulated at the protein- or RNA level, we wondered whether the subcellular localization of the gene was also related to the type of gene regulation. To test this, we split protein complex and non-complex genes into the following subcellular location/organelle groups: nucleus, nucleoli, cytoplasm, organelles, vesicles, Golgi apparatus, peroxisomes, lysosomes, ER, ribosome, proteasome, mitochondria, and plasma membrane (PM) (*Supplementary file 3L*; *Thul et al., 2017*, see 'Methods'). We then used the DNA–RNA and RNA–protein correlations of genes in the same subcellular groups (calculated at the pan-cancer level) to determine whether the cellular localization was associated with different degree of RNA-level or protein-level regulation (*Figure 3—figure supplement 2*). We asked this question separately for complex and non-complex genes. Four subcellular locations/organelles stood out as different from the rest: ribosome, proteasome, mitochondria, and PM. In fact, protein complex genes encoding subunits of ribosome, proteasome and mitochondria showed a significantly lower RNA–protein correlation (suggesting stronger protein-level regulation) compared to protein complex genes in all cell compartments (FDR < 0.001 for ribosome; FDR = 0.002 for proteasome and FDR = 0.002 for mitochondria; *Figure 3E*, *Figure 3—figure supplement 2*). No difference was observed for non-complex genes of mitochondria compared to non-complex genes in all cell compartments. Interestingly, the opposite behavior was observed for genes encoding for proteins located on the PM, and this was true for both complex and non-complex genes. Protein complex or non-complex genes encoding for PM proteins showed higher RNA–protein correlation and lower DNA–RNA correlation compared to protein complex or non-complex genes encoding for proteins at other cell compartments, respectively, suggesting a significantly lower regulation at the protein level (FDR < 0.001 for complex or non-complex genes; *Figure 3E*, *Figure 3—figure supplement 2*). This suggests that PM genes have a profoundly different type of regulation compared to other cellular locations, showing a low level of regulation at the protein level and a higher level of regulation at the RNA level (see 'Discussion'). This is consistent with the GO enrichment analysis shown in *Figure 3—figure supplement 1A* where cell structure and cell adhesion pathways were enriched in group 2. Finally, GO enrichment analysis within PM genes of a low protein-level regulation showed an enrichment of cell substrate adhesion (FDR = 1.5E-20) and cell leading edge (FDR = 8.1E-34) including *ACTN1*, *CTNND1*, *DAG1*, and others (*Supplementary file 3M*). Analysis within individual tumor types confirmed these results for the vast majority of cancer types (*Supplementary file 3N–T*).

We next asked whether the genes of distinct regulation, which we found based on the bulk RNAseq and mass spectrometry data, also show different regulation at single-cell level. We assayed the level of variability in the RNA counts across individual cells by using VarID (*Grün, 2020*), a computational method that quantifies gene expression variability locally in cell state space. We analyzed single-cell RNAseq data from six patients with colorectal cancer (CRC) (*Lee et al., 2020*). Our analysis shows that Group 2 genes (low DNA–RNA correlation and high RNA–protein correlation), preferentially regulated at the RNA level, tend to have higher expression variability than the Group 1 genes (high DNA–RNA correlation and low RNA–protein correlation) that are predominantly regulated on the protein level (*Figure 3F*).

## Protein-level changes associated with high levels of aneuploidy in primary tumors

While many studies have investigated the transcriptional changes associated with high level of aneuploidy in cancer, little is known about how these changes translate to the protein level, especially in primary tumors (*Weinstein et al., 2013*; *Rodriguez et al., 2021*). Given our finding on gene

regulation across genes and pathways (*Figure 3*), it is likely that the dysregulation of certain pathways in cancer may be overlooked by investigating exclusively changes at RNA level, as most studies have done. Thus, we set out to investigate which pathways are enriched (increased expression) or depleted (decreased expression) in highly aneuploid tumors compared to tumors with low aneuploidy both at the protein level and at the RNA level (*Figure 4A*). We note here that the goal is to identify expression changes resulting from higher aneuploidy independent of the specific chromosomes that are gained or lost. We first determined the aneuploidy score (i.e., overall aneuploidy level) for each primary CPTAC tumor by calculating the total number of chromosome arms gained or lost across all chromosomes. Second, we used a linear regression model to study the association between the RNA or protein level of each gene and the aneuploidy score. Genes were ranked based on the t-value associated to the aneuploidy score (*Supplementary file 4A and B*), and GSEA was performed on the ranked gene lists to assess which pathways were differentially expressed according to the aneuploid score. Finally, we repeated the linear modeling with confounding variables such as tumor purity and cell cycle score to get rid of their interference. For the pan-cancer analysis, we also included the tumor type as a covariate.

Pan-cancer analysis revealed several pathways to be enriched at the protein level in highly aneuploid tumors (*Figure 4B*, *Supplementary file 4C*). They included pathways related to DNA and chromatin such as DNA replication (GO: DNA replication, NES = 2.78, FDR < 0.001) and chromatin organization (GO: Chromatin organization, NES = 2.36, FDR < 0.001), RNA production and processing such as transcription elongation (GO: DNA-templated transcription elongation, NES = 2.35, FDR < 0.001), termination (GO: DNA-templated transcription termination, NES = 2.37, FDR < 0.001), RNA splicing (GO: RNA splicing, NES = 2.93, FDR < 0.001), RNA polyadenylation (GO: RNA polyadenylation, NES = 2.33, FDR < 0.001), and RNA transport (GO: mRNA transport, NES = 2.35, FDR < 0.001), pathways related to protein translation including rRNA (GO: rRNA metabolic process, NES = 2.72, FDR < 0.001), tRNA processing (GO: tRNA metabolic process, NES = 2.47, FDR < 0.001), and ribosome biogenesis (GO: Ribosome biogenesis, NES = 2.85, FDR < 0.001), and pathways related to mitochondrial gene expression (GO: Mitochondrial translation, NES = 3.49, FDR < 0.001) and transport (GO: Mitochondrial transport, NES = 2.35, FDR < 0.001) (*Figure 4B and C*, *Supplementary file 4C*). On the other hand, several pathways were depleted in highly aneuploid tumors such as cytoskeleton (GO: Actin filament organization, NES = −2.54, FDR < 0.001), cell adhesion (GO: Cell matrix adhesion, NES = −2.30, FDR = 1E-04), and pathways related to immune responses (GO: Activation of immune response, NES = −2.57, FDR < 0.001) (*Figure 4B*, *Supplementary file 4C*), consistent with previous studies (*Davoli et al., 2017*). Importantly, these results were confirmed after including additional covariates in the model (*Figure 4B*, *Supplementary file 4C*). First, we included tumor purity, which was estimated using two independent methods (nuclei percentage or using the algorithm Estimate; *Yoshihara et al., 2013*), confirming that the results were independent of the immune and stromal component of the tumor samples. Second, we included the DNA copy number change for each gene in order to assess whether the change at the protein level associated with aneuploidy was due to the fact the genes are gained or lost at the DNA level or instead to transcriptional/translational programs that are established in high aneuploid cells. The result suggested that the enriched or depleted pathways in highly aneuploid tumors were activated/suppressed as a consequence of aneuploidy. Furthermore, as gene sets related to transcription and translation also include many ribosome, rRNA, and tRNA genes of mitochondria, we removed all mitochondrial genes (mitochondrial genes encoded by the nuclear or mitochondrial DNA) before GSEA to exclude the possibility that these transcription and translation pathways were enriched only because of mitochondrial genes. The result of this analysis validated once again that transcription and translation of nuclear DNA-encoded genes are upregulated in high aneuploid cancers. Finally, since our analysis found the cell cycle pathway as one of the enriched pathways at the protein level in high aneuploidy tumors (GO: Cell cycle checkpoint, NES = 1.83, FDR = 0.01), consistent with previous findings (*Carter et al., 2006*; *Davoli et al., 2017*), we repeated the linear model including cell cycle score (*Davoli et al., 2017*; see 'Methods') to assess changes in pathways independently of cell cycle change. This model suggested that the results observed in the original model were independent of cell cycle score. Analyses of individual tumor types were generally consistent with these results obtained in the pan-cancer dataset (*Figure 4D*).

Interestingly, when we repeated the same analyses using the RNA-level expression of the same set of genes, we observed a similar trend as the enrichment at the protein level (*Figure 4B and D*,

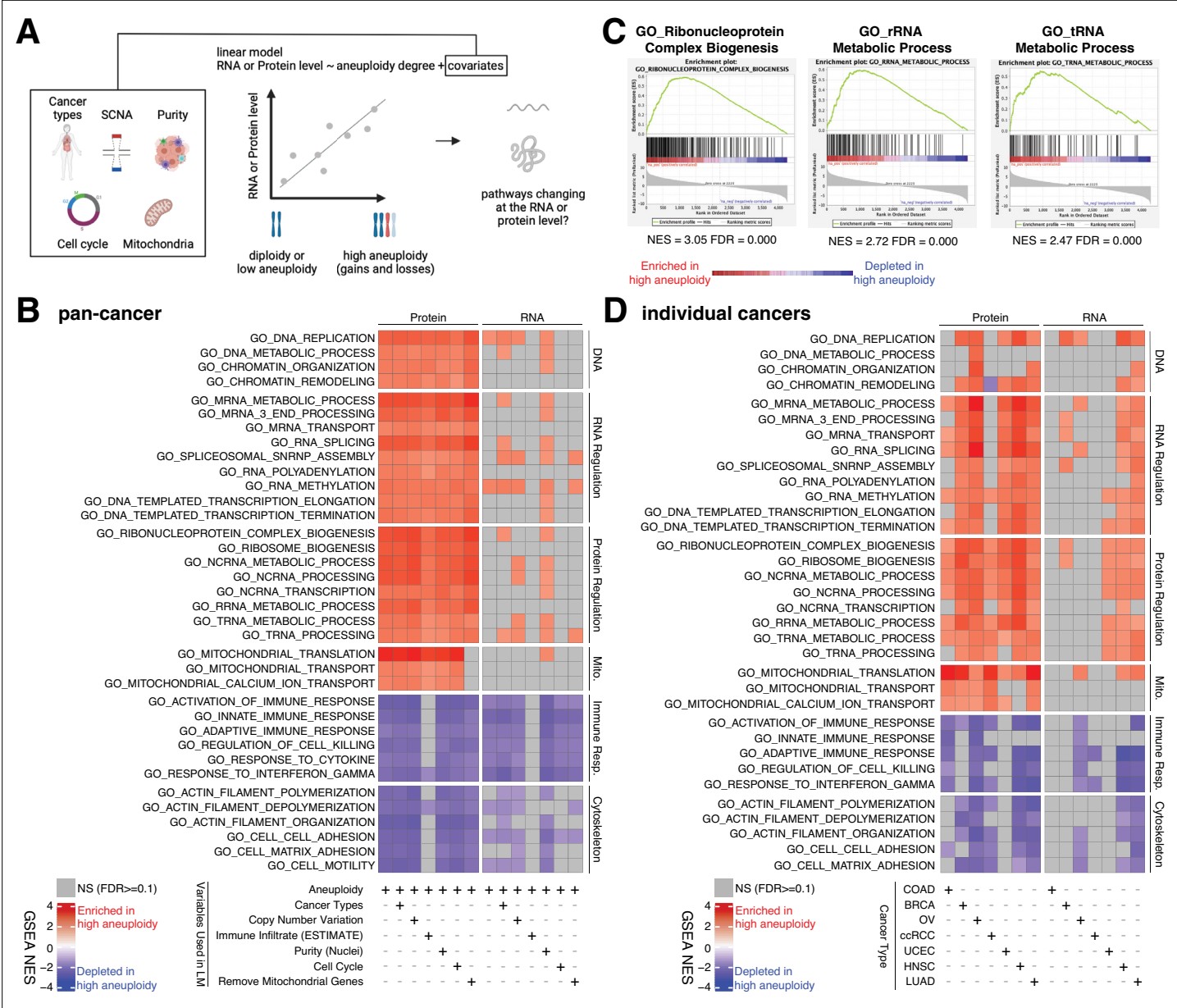

**Figure 4.** Analysis of pathways dysregulated at the RNA and protein levels in high aneuploidy tumor samples. (**A**) Schematic of the method used to identify pathways changing at the RNA and protein levels in samples of high aneuploidy. The aneuploidy degree of primary tumors (Clinical Proteomic Tumor Analysis Consortium [CPTAC]) was used to fit the RNA or protein level of each gene by linear models. Several covariates were included in the model one by one, including cancer type, gene-level copy number variation, purity, cell cycle, and mitochondria. t-values associated to the aneuploidy score were used to rank genes for Gene Set Enrichment Analysis (GSEA). (**B**) A heatmap showing the enrichment score for the indicated pathways significantly enriched (red) or depleted (blue) in high versus low aneuploidy tumor samples. Specific gene sets related to DNA, RNA, and protein regulation and mitochondria are enriched, and those related to immune response and cytoskeleton are depleted at the protein level in aneuploid tumor tissues. Covariates were included in the model to control for cancer types, gene-level copy number variation, purity, and cell cycle scores. Mitochondrial genes were removed in the last column. The gene sets whose false discovery rate (FDR) are larger than 0.1 are shown in gray. (**C**) Enrichment plots of three pathways related to protein translation in tumor tissues at the protein level: ribonucleoprotein complex biogenesis, rRNA metabolic process, and tRNA metabolic process. The normalized enrichment scores and FDR are shown below the corresponding enrichment plots. (**D**) A heatmap showing the enrichment of the same gene sets as (**B**) in individual cancer types. RNA or protein expression for each gene was fit by the aneuploidy degree without the inclusion of other covariates. Gene sets enriched in high aneuploid samples are in red while those depleted in high aneuploid samples are in blue. The gene sets whose FDR are larger than 0.1 are shown in gray.

*Supplementary file 4C and D*). However, the enrichment of some pathways (DNA, RNA, and protein regulation and mitochondrial translation and transport) at RNA level is weaker compared to those at protein level (more pathways were not significant at the RNA level compared to the protein level in *Figure 4B and D*). For example, at the RNA level, only one pathway (DNA replication pathway) was significantly altered, while at the protein level, 24 pathways were found to be enriched (pan-cancer analysis, *Figure 4B*). This difference between transcriptome and proteome is consistent with our findings in *Figure 3C* that pathways related to transcription, translation, and mitochondria are preferentially regulated at the protein level.

Altogether, these results suggest that tumors with high degree of aneuploidy show enrichment in pathways related to protein translation, mitochondria, and RNA processing, and depletion of pathways related to immune-related response, which are independent of other covariates such as purity and cell cycle score. In most tumors studied these changes are much more evident at the protein level than at the RNA level, suggesting that their upregulation is due at least in part to a protein-level regulation.

## Discussion

How cells control the abundance of their proteins in physiological and pathological conditions is a fundamental question. Both the regulation at the RNA and protein level can contribute to the protein abundances. However, the relative contribution of these two layers of regulation remains unclear (*Vogel and Marcotte, 2012*). Furthermore, it remains unknown whether the relative contribution of the RNA- and protein-level regulation varies based on the DNA copy number, interaction with other proteins, protein function and location, and so on. In this study, our proteogenomic analysis allowed us to uncover general principles linking mainly gene function to the mechanism of gene regulation. In particular, we found that the genes and pathways that have stronger protein-level regulation tend to have weaker RNA-level regulation and vice versa, suggesting that each pathway has a predominant type of regulation. Specifically, certain pathways including protein translation, protein folding, mRNA processing, and cellular respiration tend to have a strong protein-level regulation while other pathways such as cell adhesion and chemotaxis tend to have a strong RNA-level regulation (*Figure 5*).

### Tissue specificity of RNA- or protein-level compensation

Pan-cancer analysis revealed several forms of gene compensation that are common across the majority of tissue types (*Figure 1B*). We found that strong compensation at the protein level is common among the seven tumors studied here, while compensation at the RNA level is less common and showed heterogeneous tissue-specific patterns. The protein-level compensation is stronger for genes in protein complexes than non-complex genes, consistent with previous reports (*Stingele et al., 2012*; *Torres et al., 2010*). Interestingly, the existence of protein-level compensation and its higher degree for protein complex genes were true not only for DNA gains, but also for DNA losses. Consistent with our findings, a recent study reported protein-level compensation after chromosome loss (*Chunduri et al., 2021*), although in this study no significant difference was reported between complex and non-complex genes, perhaps due in part to the limited number of genes on the lost chromosomes. The protein compensation of DNA gains for complex genes is thought to occur through degradation of the overabundant protein subunits (*McShane et al., 2016*). In principle, this model could also explain protein compensation after DNA loss and why compensation is stronger for protein complexes. Some protein complex subunits are more likely to be overproduced and degraded soon after translation, leading to an adjustment of their level. Protein complex genes that are lost could be compensated for by decreased protein degradation after overproduction (*McShane et al., 2016*). Future studies are needed to shed light on this process.

Individual tumor types showed unexpected tissue specificities for type and degree of compensation (*Figure 1C and D*). For example, while for protein-level compensation, six of the seven tumor types studied showed evidence of protein-level compensation that was stronger at the SCNA extremes, e.g., for deep losses and high gains (*Figure 1D*), in breast cancer protein compensation for losses was observed only for complex proteins (*Figure 1C*). Lung adenocarcinoma did not show any compensation, either at the protein or at the RNA level. UCEC and HNSC showed similar patterns and degree of

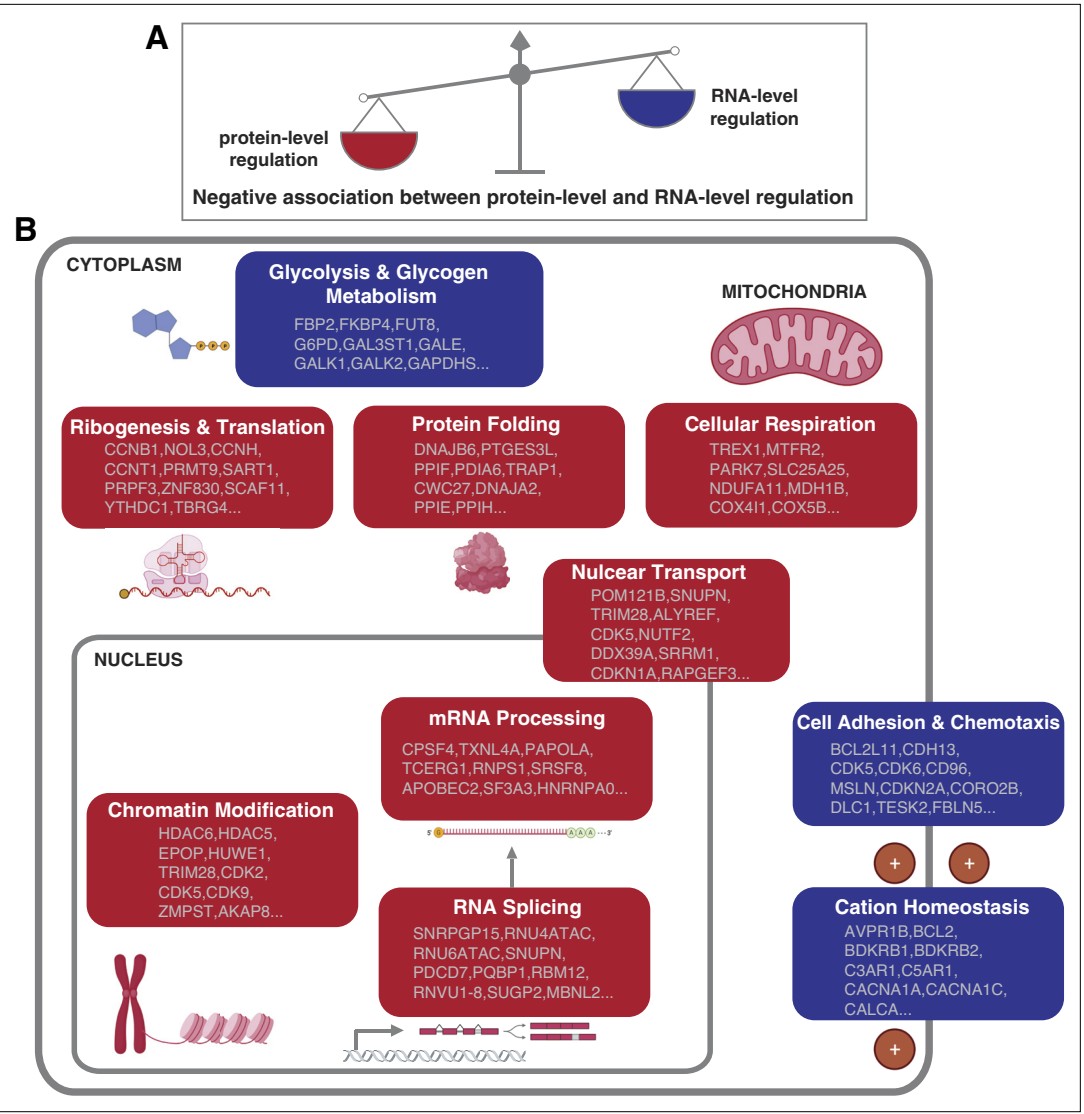

**Figure 5.** Negative association between RNA- and protein-level regulation across cellular pathways.
(**A**) Schematics representing the negative correlation between protein-level regulation and RNA-level regulation across pathways (see also *Figure 3B*). (**B**) Schematics of representative cellular pathways showing a preferential regulation at the RNA level (blue) or protein level (red). For each pathway, approximately 10 representative genes are shown. See also *Figure 3*.

gene compensation, limited to protein-level compensation. RNA-level compensation was observed in four tumors, and exhibited far more variable tissue-specific patterns. Renal cancer and breast cancer showed RNA-level compensation for deep losses and high gains, respectively. Furthermore, RNA-level compensation both for losses and gains was observed for colon and ovarian cancer, the latter for non-comple only (see also below). To our knowledge, this is the first study to investigate and report tissue-specific RNA- and protein-level compensation across different tumor types.

## Negative association between protein-level and RNA-level regulation across genes and pathways: Regulation tends to occur either at the RNA or protein level

In this study, we used the DNA–RNA correlation and RNA–protein correlation to estimate the degree of RNA-level and protein-level regulation. We observed that genes with similar pattern of regulation tended to be enriched in functional pathways thus to perform related functions (*Figure 3—figure supplement 1*). For example, genes implicated in translation and RNA processing tended to have

stronger protein-level regulation and weaker RNA-level regulation while genes functioning in cell structure and adhesion tended to have lower protein-level regulation and stronger RNA-level regulation (Group 1 and 2 analyses, *Figure 3—figure supplement 1*). This indicates that genes sharing similar biological functions may have evolved similar types of regulation. Interestingly, a previous study investigating the RNA and protein half-lives reported functional similarities among genes and pathways with similar RNA and/or protein half-lives (*Schwanhäusser et al., 2011*).

Strikingly we observed a significant negative correlation between the RNA-level and protein-level regulation across both genes (*Figure 3A*) and cellular pathways (*Figure 3B*, *Figure 5*), both in pan-cancer analysis and in individual tumor types (*Supplementary file 3K*). This finding held true if we used normal tissue datasets to calculate the RNA–protein correlations (*Figure 3C*). This suggests that the degree of RNA-level regulation tends to be inversely associated with the degree of protein-level regulation and that this is not restricted to aneuploid cancer cells but is true also in normal cells. One possible explanation is that for certain genes protein-level regulation may be difficult or impossible, leaving RNA-level regulation as the only feasible gene regulation mechanism. For example, for proteins involved in cell adhesion (even those involved in protein complexes), it may be difficult to degrade them once transported to the location where they normally function. Thus, in this case, strong RNA-level regulation may be more effective than a post-translational regulation mechanism. On the other hand, for cytoplasmic protein complex genes, it may be not possible to achieve a strong RNA-level regulation and thus they have to be regulated at post-translational level, such as by protein degradation or by co-regulating protein synthesis of different subunits (*Kamenova et al., 2019*; *Shiber et al., 2018*; *Taggart et al., 2020*; *Taggart and Li, 2018*). Although it could be more energetically favorable to regulate a gene at the RNA level compared to the protein level (*Franks et al., 2017*; *Wagner, 2005*), it is likely difficult to regulate at the RNA level for large mammalian protein complexes whose subunits are scattered around the eukaryotic genome (in contrast to bacterial operons) (*Buccitelli and Selbach, 2020*). An additional possibility may be related to the cellular localization of the proteins. For example, genes encoding mitochondrial proteins have a strong protein-level regulation; since these proteins are synthetized before import into mitochondria (*Isaac et al., 2018*), regulation of protein function and complex assembly has to occur at the protein level within the organelle.

Furthermore, the distinct patterns of gene expression found from the bulk RNAseq and mass spectrometry experiments also impact the variability in gene expression at single-cell level. We found that genes with stronger regulation at the RNA level tend to have higher expression variability across individual cells (*Figure 3F*). This observation suggests that regulation on the RNA level leads to increased cell-to-cell variability of the number of RNA molecules, whereas reduced regulation of RNA levels implies robustness of RNA output. Hence, a potential for strong regulation on the RNA level comes at the cost of increased cell-to-cell variability, likely due to the requirement of an increased number of stochastic gene regulatory interactions.

## Types of gene regulation and other gene features: Cellular localization and mRNA half-life

Protein localization was a significant predictor of the type of gene regulation. Consistent with previous observations (*Taggart et al., 2020*), ribosome and proteasome complexes showed the strongest level of protein-level regulation. As mentioned above, mitochondrial genes belonging to protein complexes showed a similarly strong protein-level regulation. On the contrary and to our surprise, proteins that reside on the PM showed a weaker protein-level regulation and a stronger RNA-level regulation compared to other cell compartments. While we cannot exclude that this may be due to technical difficulties in detecting membrane proteins, if this was the case, we would perhaps expect the RNA–protein correlation to be lower than for other complex genes (*Figure 3E*). While misfolding-induced degradation of proteins in the cytosol or ER (e.g., through the unfolded protein response) is well understood, little is known about the consequence of misfolding or mis-assembly for proteins on the PM (*Hetz et al., 2020*).

We also noticed an interesting association with RNA half-life. RNA half-life was positively associated with the RNA–protein correlation (rho = 0.508, p=0.001) and negatively associated with the DNA–RNA correlation (rho = −0.516, p=0.002) (*Supplementary file 2H*). In other words, pathways with a strong protein-level regulation (Group 1, *Figure 3—figure supplement 1*) tended to have a low RNA half-life and pathways with a strong RNA-level regulation (Group 2, *Figure 3—figure*

*supplement 1*) tended to have a high RNA half-life. Since pathways that tended to be strongly regulated at the RNA level have long lived RNA, this suggests that most regulation is at the transcriptional level, not the RNA degradation level.

## Pathways dysregulated in aneuploid cancers at the protein level

We observed that among the pathways significantly upregulated in high versus low aneuploid tumors at the protein level (both pan-cancer and individual tumor-type analyses), there were pathways related to RNA transcription, processing, transport and regulation, tRNA and ribosome biogenesis, and protein synthesis and translation. However, the change of those pathways at the RNA level is less significant. This is consistent with our finding that these gene sets tend to have stronger protein-level regulation. Interestingly, in the flagship endometrial CPTAC study, ribosome biogenesis was one of the most strongly enriched pathways in the serous uterine cancer subtype (*Dou et al., 2020*), which is the one that shows the highest level of aneuploidy among all uterine cancer subtypes. We also note that, based on recent studies, most of these pathways that we found upregulated in primary tumors were not significantly enriched in high aneuploid cancer cell lines, based on recent reports (*Schukken and Sheltzer, 2021*). This suggests that the tumor microenvironment may play an important role in shaping the level of these pathways.

## Open questions

An outstanding question remains about the mechanism of protein-level compensation and regulation. Previous studies suggest that the regulation occurs at the level of protein degradation (*Dephoure et al., 2014*; *Torres et al., 2010*). However, it seems now clear that protein degradation coexists with regulation at the protein synthesis level, and that at least for certain complexes, the vast majority of the protein-level regulation occurs at the protein synthesis level with fine-tuning happening through protein degradation (*Kamenova et al., 2019*; *Shiber et al., 2018*; *Taggart et al., 2020*; *Taggart and Li, 2018*). Additional studies are needed to better characterize the level of translation or proteasome regulation across cell compartment, protein complexes, and cellular pathways.

## Methods

### Datasets

All CPTAC-related SCNA, RNA, protein, and mutation data were obtained from the CPTAC portal (https://cptac-data-portal.georgetown.edu/datasets) or from CPTAC. The number of patients and genes of individual cancers used in our analyses are listed in *Supplementary file 1A*.

DNA copy number was obtained for samples from the CPTAC analysis of TCGA samples from COAD, BRCA, and OV via Affymetrix SNP 6.0 (SNP6) as described previously (*The Cancer Genome Atlas Network, 2012*). For the independent CPTAC cohorts for COAD, BRCA, and OV, DNA copy number was derived from WES as described previously (*Krug et al., 2020*; *McDermott et al., 2020*; *Vasaikar et al., 2019*). Samples from the CPTAC cohorts for ccRCC, UCEC, HNSC, and LUAD were processed using WES and WGS as described previously (*Clark et al., 2019*; *Dou et al., 2020*; *Gillette et al., 2020*; *Huang et al., 2021*).

RNA-sequencing from the CPTAC samples obtained from TCGA (COAD, BRCA, and OV) was achieved by aligning reads to the human genome (hg19) using the BWA algorithm (http://bio-bwa. sourceforge.net/) as described previously (*The Cancer Genome Atlas Network, 2012*). Independent datasets for CPTAC COAD, BRCA, and OV were processed as described previously (*Krug et al., 2020*; *McDermott et al., 2020*; *Vasaikar et al., 2019*). CPTAC cohorts ccRCC, UCEC, HNSC, and LUAD were processed as described previously (*Clark et al., 2019*; *Dou et al., 2020*; *Gillette et al., 2020*; *Huang et al., 2021*).

For BRCA and LUAD, the Spectrum Mill software package v7.0 pre-release (Agilent Technologies, Santa Clara, CA) was used for MS data analysis. Protein identification was performed by searching the MS/MS spectra against protein sequence database obtained using the UCSC Table Browser (https://genome.ucsc.edu/cgi-bin/hgTables) on September 14, 2016, that contains 37,579 proteins mapped to the human reference genome (hg19), adding common contaminants, mitochondrial proteins, and non-canonical small open-reading frames. The searches were performed allowing ±20 ppm mass tolerance for precursor and product ions, allowing for common modifications. Peptide spectrum matches

(PSMs) were filtered for 30% minimum matched peak intensity and target-decoy-based FDR estimates at the PSM level, and for proteins protein level for each TMT-plex for all TMT-plexes for a tumor type, and for phospho all TMT-plexes for a tumor type, and for phosphorylation at the site levels. Normalization of each peptide was performed using the common reference, and a two-component Gaussian mixture model-based normalization was used to nullify the effect of differential protein loading and/ or systematic MS variation.

For COAD, OV, and UCEC, MS-GF+ v9881 (*Gibbons et al., 2015*; *Kim et al., 2008*; *Kim and Pevzner, 2014*) was used to search against the RefSeq human protein sequence database downloaded on June 29, 2018 (hg38; 41,734 proteins), combined with 264 contaminants (e.g., trypsin, keratin) using partial tryptic peptides, ±10 ppm parent and fragment ion tolerance, allowing for isotopic error in precursor ion selection and common modifications (static carbamidomethylation [+57.0215 Da] on Cys residues and TMT modification [+229.1629 Da] on the peptide N terminus and Lys residues, and dynamic oxidation [+15.9949 Da] on Met residues), and including decoy sequences generated by reversing the protein sequences. Peptides were filtered using a maximum FDR of 1% at peptide level using PepQValue < 0.005 and parent ion mass deviation < 7 ppm criteria. A minimum of six unique peptides per 1000 amino acids of protein length was required for achieving 1% at the protein level within the full dataset. The TMT reporter ion intensities were extracted using MASIC (*Monroe et al., 2008*). Relative protein levels were calculated as the ratio of sample abundance to reference abundance using the summed reporter ion intensities from peptides that could be uniquely mapped to a gene. The relative abundances were log2 transformed and zero-centered for each gene to obtain final relative abundance values. Each sample was median centered to adjust for differences in laboratory conditions and sample handling.

For HNSC and ccRCC, the MSFragger version 3.0 (*Kong et al., 2017*) was used to search the RefSeq human protein sequences and an equal number of decoy sequences using tryptic and semi-tryptic peptides allowing two missed cleavages, a mass tolerance of 10 ppm, and allowing isotope errors, mass calibration, spectral deisotoping, and parameter optimization (*Yu et al., 2020*). Cysteine carbamidomethylation, lysine and peptide N-terminal TMT labeling were specified as fixed modifications, and methionine oxidation and serine TMT labeling were specified as variable modifications, and for the phosphopeptide-enriched data, phosphorylation of serine, threonine, and tyrosine residues was allowed. Philosopher toolkit version v3.2.8 (*da Veiga Leprevost et al., 2020*) was used for postprocessing. The protein groups assembled by ProteinProphet (*Nesvizhskii et al., 2003*) were filtered to 1% protein-level FDR. To generate summary reports, TMT-Integrator (*Djomehri et al., 2020*) was used. PSM mapping to common contaminant proteins was excluded, and both unique and razor peptides were used for quantification. The reporter ion intensities of each PSM were log2 transformed and normalized by the reference channel intensity median centered after removal of outliers. For HNSC, we specifically focused on the HPV-negative HNSC, because compared with HPV-positive HNSC, the lethal subtype has very distinct SCNA profiles, patterns and interactions with cell cycle and immune signaling pathways (*William et al., 2021*).

The list of protein complex genes (Core complexes) was downloaded from the CORUM database v3 (*Ruepp et al., 2008*). All CCLE-related SCNA, RNA, and protein data of cancer cell lines were collected from DepMap (19Q4). NCI-60 data was downloaded from CellMiner (2.8.1). The mRNA and protein expression data of 29 human tissues was from *Wang et al., 2019*. GTEx RNA data was downloaded from GTEx portal (v8, https://gtexportal.org/home/), and the corresponding protein data was downloaded from *Jiang et al., 2020* scRNAseq of CRC patients (*Lee et al., 2020*) was retrieved from Gene Expression Omnibus (GSE132465).

## Calculation of log2FC for DNA, RNA, and protein values

Before starting the calculation, low-expression genes whose RNA level were within the bottom 10% in individual tumor tissues were removed. Only genes that had DNA, RNA, and protein data were kept for the following analyses. For each gene of each cancer (80–110 patients for each cancer), we defined the patients that do not have a DNA copy number change (log2 copy number ratio is between –0.2 and 0.2) as the neutral group. We considered the RNA and protein expression median of this group as the *neutral* RNA or protein level. Then we calculated the log2FC at the DNA, RNA, and protein level for each gene in each sample compared to the *neutral* DNA, RNA, or protein level. For each gene in each sample, we determined whether there is a DNA loss (DNA log2FC is between –0.65

and –0.2), deep loss (DNA log2FC < –0.65), gain (DNA log2FC is between 0.2 and 0.65), or high gain (DNA log2FC > 0.65). For the pan-cancer analysis, the log2FC data of individual cancers were pooled together (682 patients in total). Quality control was done using principal component analysis on the pooled log2FC data, confirming that no cancer type was distinct from others. To calculate the log2FC of cancer cell lines (CCLE), the cancer types of more than 13 cell lines were used (284 samples from 11 cancer types). As for CPTAC, the log2FC of DNA, RNA, and protein were calculated for each gene in each cancer. Then the log2FC values of different cancers were merged.

## Calculation of the compensation score

In order to quantify the degree of RNA- or protein-level compensation, we calculated a CS for each gene in each sample determined as the difference between the RNA or protein log2FC and the DNA log2FC as shown in the following formula. CS is larger than 0 when compensation exists. A higher CS means higher compensation.

$$\text{compensation score}\,(CS) = \begin{cases} DNA\ log2FC - RNA\ or\ protein\ log2FC\ (when\ DNA\ log2FC > 0) \\ RNA\ or\ protein\ log2FC - DNA\ log2FC\ (when\ DNA\ log2FC < 0) \end{cases}$$

To test whether there was significant compensation in each group of DNA change, we used bootstrapping method by randomly sampling the CS of genes in the specific groups 10,000 times and calculated the median of CS for each time by boot package (v1.3-28). 95% confidential interval of CS was calculated by the basic method of boot.ci function. The p-value was calculated at one-tail to test the null hypothesis (the CS is not larger than 0), which was corrected by FDR method. To compare whether there was significant difference between CS of protein complex genes and non-complex genes in the specific groups, the CS was randomly resampled 10,000 times and the difference of CS was calculated for each time by boot package. 95% confidential interval of CS difference (CS for protein complex genes – CS for non-complex genes) was calculated by the basic method of boot.ci function (positive values mean stronger compensation for protein complex genes). The p-value was calculated at two-tail to test the null hypothesis (the CS difference equals 0), which was corrected for FDR Benjamini–Hochberg method (*Benjamini and Hochberg, 1995*).

## DNA–RNA and RNA–protein correlation for each gene

Only the genes that had DNA, RNA, and protein data were considered for these analyses. For each cancer type, genes that showed no or very little change at the DNA level (log2 copy number ratio is between –0.02 and 0.02) in more than 70% of the patients were removed because those genes are likely to influence the correlations analyses. The analyses were also confirmed using all genes. For each gene, we then calculated the DNA–RNA and RNA–protein Spearman's correlation (rho value). Next, we merged the correlation of different tumor types at the gene level. More specifically, for each gene, we calculated the mean of correlation coefficients across different tumors and considered this value as the correlation coefficient for the pan-cancer. The same method was applied to CCLE and NCI-60 datasets and to normal tissues datasets (*Alley et al., 1988*; *Barretina et al., 2012*), The RNA–protein Spearman's correlation (rho value) for each gene was calculated by the same method. The p-value was evaluated based on a 10,000-times bootstrapping test to compare the median difference between CORUM complex genes with NoCORUM genes. All the p-values were adjusted by FDR using the Benjamin–Hochberg method (*Benjamini and Hochberg, 1995*).

## Bootstrapping strategy

A bootstrapping strategy was used to identify the difference between two groups or between one group with a certain value. For example, to compare protein complex and non-complex genes, this procedure generated 10,000 randomly resampled datasets from the whole complete gene set with replacement: $\left(X^i, Y^i\right)_{i=1,2,\ldots,10000}$, where X and Y would be assigned as new complex and new non-complex genes, respectively, each time. Then for each resampled dataset, we calculated the median of complex and non-complex genes. A distribution was built based on the 10,000 resampled medians. Finally, we compared the median distribution with the 'real' median to calculate the p-value. The result

of bootstrapping test was also confirmed by Mann–Whitney $U$ test and Kolmogorov–Smirnov test (*Supplementary file 2K*).

## Gene-level and pathway-level analysis of the DNA–RNA and RNA–protein correlations

To estimate the association between the DNA–RNA (DR) and RNA–protein (RP) correlations at the gene level (*Figure 3A*), we first calculated the pan-cancer DR and RP Spearman's correlations (rho values) for each gene, resulting in a density distribution f(DR, RP). We split the DR range into a series of windows (40 bins), and in each of the windows, *i*, the RP value of the maximum density, $RP_i$, was chosen to represent the RNA–protein correlation of genes in the windows, that is,

$$RP_i = \underset{RP}{argmaxf}\left(DR_i, RP\right)$$

Therefore, a series of representative points were determined: ($DR_i$,$RP_i$), *i* = 1, 2, …, 40. The slope and the rho (Spearman's correlation coefficient) of those representative points were used as an estimate of the association between DNA–RNA and RNA–protein correlations.

For the enrichment and single-cell analysis of genes of distinct regulation, genes were divided into two groups: Group 1, composed of genes with a high DNA–RNA correlation (top 35%, rho > 0.43) and a low RNA–protein correlation (bottom 35%, rho < 0.31), and Group 2 of genes with a low DNA–RNA correlation (bottom 35%, rho < 0.24) and a high RNA–protein correlation (top 35%, rho > 0.50). GO enrichment analysis was then used to test whether these genes showed enrichment or not for different pathways (msigdbr, v7.4.1, category = C5). The single-cell analysis is discussed below.

For the pathway-level analysis, we considered the cellular pathways utilized in the previous study as they represent most cellular functions (*Schwanhäusser et al., 2011*). The genes of each pathway were identified by the msigdb GSEA database (v7.4). For each pathway, the median of the rho values across the genes in the pathway was used as the correlation value associated to the pathway (e.g., the median of DNA–RNA rho correlation values for the genes in the cell cycle pathway would represent the DNA–RNA correlation value for cell cycle pathway).

## Phylogenetic conservation analysis

phyloP scores (hg19.100way.phyloP100way.bw; *Hubisz et al., 2011*) were downloaded from UCSC (http://hgdownload.cse.ucsc.edu/goldenpath/hg19/phyloP100way/, positive value: more conserved; negative value: less conserved). Genome-related information was downloaded from GENCODE (v19). The median of phyloP scores at all coordinates of the same genes was used as the scores at the gene level. During these analyses, genes with top 30% and bottom 30% of the phyloP score were picked for the further analysis.

## Subcellular location analysis

Subcellular location data was downloaded from The Human Protein Atlas (*Uhlén et al., 2015*). The 'Main location' data was used for the Subcellular location analysis. Subcellular locations included nucleus (including nucleoplasm, nuclear speckles, nuclear bodies, and nuclear membrane), cytoplasm (including microtubules, cytosol, actin filaments, centrosome, centriolar satellite, cytoplasmic bodies, intermediate filaments, cytokinetic bridge, mitotic spindle, and microtubule ends), nucleoli (including nucleoli, nucleoli fibrillar center, and nucleoli rim), mitochondria, ER, PM, proteasome, and ribosome. For each subcellular location, we calculated the DNA–RNA and RNA–protein Spearman's correlation for the genes in each subcellular location. The p-value was evaluated based on a 10,000 times bootstrapping test. All the p-values were adjusted by FDR method.

## Generate single-cell-derived hCEC clones containing aneuploidy

To derive a panel of isogenic aneuploid cell lines, hTERT-immortalized TP53-KO (non-tumorigenic) hCEC cells (derived from hCEC cells *Roig et al., 2010*) after treatment with a sgRNA taregting *TP53* (*Sack et al., 2018*) were treated with reversine (0.2 µM for 24 hr), an MPS1 inhibitor that prevents correct chromosome attachment and spindle checkpoint to induce random chromosome missegregation (*Santaguida et al., 2015*). Then the cells were plated at a low density and grew until the colonies

formed. Those single-cell-derived clones were picked using glass cylinders. To identify the levels and patterns of aneuploidy, the clones were sequenced by shallow WGS. The transcriptome and proteome were measured by RNA-sequencing and mass spectrometry (see below).

## Shallow whole-genome sequencing

hCEC clones were plated in 48-well plates 1 day before the collection. At the second day, genomic DNA was extracted from trypsinized cells using 0.3 µg/µL Proteinase K (QIAGEN #19131) in 10 mM Tris pH 8.0 for 1 hr at 55°C, then heat-inactivated at 70°C for 10 min. DNA was digested using NEBNext dsDNA Fragmentase (NEB #M0348S) for 25 min at 37°C followed by magnetic DNA bead cleanup with Sera-Mag Select Beads (Cytiva #29343045), 2:1 bead to lysate ratio by volume. We created DNA libraries with an average library size of 320 bp using the NEBNext Ultra II DNA Library Prep Kit for Illumina (NEB #E7103) according to the manufacturer's instructions. Quantification was performed using a Qubit 2.0 fluorometer (Invitrogen #Q32866) and the Qubit dsDNA HS kit (#Q32854). Libraries were sequenced on an Illumina NextSeq 500 at a target depth of 4 million reads in either paired-end mode (2 × 36 cycles) or single-end mode (1 × 75 cycles). Low-pass (~0.1–0.5×) WGS reads of hCEC were aligned to reference human genome hg38 by using BWA-mem (v0.7.17) (*Li and Durbin, 2009*) and followed by duplicate removal using GATK (Genome Analysis Toolkit, v4.1.7.0) (https://gatk.broadinstitute.org/hc/en-us) to generate analysis-ready BAM files. BAM files were processed by the R Package CopywriteR (v1.18.0) (*Kuilman et al., 2015*) to call the arm-level copy numbers.

## RNA-sequencing

hCEC clones were plated in 6-well plates 1 day before the collection. On the second day, the cells were checked to make sure their confluency was within 70–90% and morphology was normal. Then the cells were washed twice in PBS and stored at –80°C immediately. Total RNA was isolated from each sample using PicoPure RNA Isolation kit (Life Technologies, Frederick, MD) including the on-column RNase-free DNase I treatment (QIAGEN, Hilden, Germany) following the manufacturer's recommendations. To purify RNA for sequencing, we used the QIAGEN RNeasy Mini Kit (QIAGEN 74106). RNA concentration and integrity were assessed using a 2100 BioAnalyzer (Agilent, Santa Clara, CA). Sequencing libraries were constructed using the TruSeq Stranded Total RNA Library Prep Gold mRNA (Illumina, San Diego, CA) with an input of 250 ng and 13-cycle final amplification. Final libraries were quantified using High Sensitivity D1000 ScreenTape on a 2200 TapeStation (Agilent) and Qubit 1x dsDNA HS Assay Kit (Invitrogen, Waltham, MA). Samples were pooled equimolar with sequencing performed on an Illumina NovaSeq6000 SP 100 Cycle Flow Cell v1.5 as paired-end 50 reads.

## RNAseq pipeline

Total RNA-sequencing reads of hCEC were mapped to the human genome hg38 by STAR (version 2.7.7a) (*Dobin et al., 2013*) using the 2-pass model. hg38 sequence and RefSeq annotation were downloaded from the UCSC table browser. RSEM (version 1.3.1) (*Li and Dewey, 2011*) was used to quantify gene and transcript expression levels. RSEM output the gene-level raw counts and fragments per kilobase of transcript per million mapped reads (FPKM) results in table format. The RNA RSEM data will be filtered for genes with median FPKM > 1 for use in downstream analyses.

## qPCR to validate RNAseq result

To validate the gene expression change calculated from RNAseq, we used qPCR to check the RNA log2FC of certain genes. hCEC clone A12 and the diploid control (D29) were used for this purpose. The preparation of cells and extraction of RNA were the same as RNAseq. Then one-step real-time RT-qPCR reactions were performed in a Lightcycler 480 instrument (Roche Diagnostics) using the One Step PrimeScript RT-PCR Kit (Perfect Real Time) (Takara Bio RR064A). Reverse transcription and probe-based qPCR reactions were performed in a single tube from 50 ng of isolated RNA as follows: one cycle of reverse transcription at 42°C for 5 min and 95°C for 10 s, then one cycle of enzyme activation at 95°C for 5 min, and lastly 45 cycles of 95°C for 5 s, and annealing at 63°C for 20 s. A single acquisition was taken after each cycle. Reactions were done in triplicates, and a non-RNA control was used. Predesigned TaqMan gene expression assays for the five selected genes and one housekeeping gene were purchased from Thermo Fisher Scientific: ABCB1 (Hs00184500_m1), CCT6A (Hs00798979_s1), PDGFRA (Hs00998018_m1), RAC1 (Hs01902432_s1), RPL9 (Hs01552541_g1), YY1

(Hs00998747_m1). The mean cycle threshold, standard deviation, delta Ct, and delta-delta Ct were calculated using Microsoft Excel. YY1 housekeeping gene was used to normalize the gene target value and SD. A comparative Ct method was used to calculate the delta-delta Ct between our test sample and the calibrator sample.

## Global protein abundance profiling

Cell pellets were lysed in the following buffer: 8 M urea, 100 mM Tris, pH = 8.5, 10 mM TCEP, and 40 mM CAA (150 µl/sample) and sonicated in probe sonicator for 1 × 5 s cycle at amplitude of 50%. Lysates were incubated for 30 min at 56°C in a thermoshaker at 1000 rpm. Insoluble debris were removed by centrifugation (5 min at 16,000 × *g*). Protein concentrations were measured by A280 method, and proteins were digested with trypsin at 50:1 (w/w) ratio at 37°C (lysates were diluted sixfold with 20 mM Tris, pH = 8 prior to digestion). Subsequently, samples were acidified with 10% FA to final of 0.5% FA and centrifuged to remove undigested material. Peptides were desalted on tC18 Waters SepPak cartridges and eluates were dried on speedvac.

50 µg of digest from each sample were resolubilized in 20 µl of 50 mM HEPES buffer pH = 8.5. 8 µl of TMTPro reagent (can stock at 12.5 mg/ml) were added, and labeling was allowed to proceed for 30 min at room temperature (RT). Excess of label was quenched by adding 40 µl of 500 mM ABC buffer (30 min at 37°C). Labeled peptides were mixed together to create 2 × 16 plex TMT batches, which were subsequently desalted on tC18 SepPak cartridges, concentrated on speedvac, and fractionated offline.

500 µg of peptides were fractionated using a Waters XBridge BEH 130A C18 3.5 um 4.63 mm ID × 250 mm column on an Agilent 1260 Infinity series HPLC system operating at a flow rate of 1 ml/min with three buffer lines: buffer A consisting of water, buffer B canACN, and buffer C of 100 mM ammonium bicarbonate. Peptides were separated by a linear gradient from 5% B to 35% B in 62 min followed by a linear increase to 60% B in 5 min, and ramped to 70% B in 3 min. Buffer C was constantly introduced throughout the gradient at 10%. Fractions were collected every 60 s. Fractions from 30 to 64 were used for LC-MS/MS analysis.

LC separation was performed online on EvosepOne LC (*Bache et al., 2018*) utilizing Dr Maisch C18 AQ, 1.9 µm beads (150 µm ID, 15 cm long, Cat# EV-1106) analytical column. Peptides were gradient eluted from the column directly to Orbitrap HFX mass spectrometer using 44 min evosep method (30SPD) at a flow rate of 220 nl/min. Mass spectrometer was operated in either data-dependent acquisition mode DDA. High-resolution full MS spectra were acquired with a resolution of 120,000, an AGC target of 3e6, with a maximum ion injection time of 100 ms, and scan range of 400–1600 m/z. Following each full MS scan, 20 data-dependent HCD MS/MS scans were acquired at the resolution of 60,000, AGC target of 5e5, maximum ion time of 100 ms, one microscan, 0.4 m/z isolation window, nce of 30, fixed first mass 100 m/z, and dynamic exclusion for 45 s. Both MS and MS2 spectra were recorded in profile mode.

## Proteome analysis pipeline

MS data were analyzed using MaxQuant software version 1.6.15.0 (*Cox and Mann, 2008*) and searched against the SwissProt subset of the human UniProt database (http://www.uniprot.org/) containing 20,430 entries. Database search was performed in Andromeda (*Cox et al., 2011*) integrated in MaxQuant environment. A list of 248 common laboratory contaminants included in MaxQuant was also added to the database as well as reversed versions of all sequences. For searching, the enzyme specificity was set to trypsin with the maximum number of missed cleavages set to 2. The precursor mass tolerance was set to 20 ppm for the first search used for nonlinear mass recalibration and then to 6 ppm for the main search. Oxidation of methionine was searched as variable modification; carbamidomethylation of cysteines was searched as a fixed modification. TMT labeling was set to lysine residues and N-terminal amino groups, and corresponding batch-specific isotopic correction factors were accounted for. The FDR for peptide, protein, and site identification was set to 1%, and the minimum peptide length was set to 6. To transfer identifications across different runs, the 'match between runs' option in MaxQuant was disabled. Only precursors with minimum precursor ion fraction (PIF) of 75% were used for protein quantification. Match between runs option was enabled and RAW TMT reporter ion intensities of peptide features were used for subsequent data analysis in MSstatsTMT (*Huang et al., 2020*).

Subsequent data analysis was performed in either Perseus (*Tyanova et al., 2016*) (http://www.perseus-framework.org/) or using R environment for statistical computing and graphics (http://www.r-project.org/).

## Quantification of aneuploidy degree

The segment files for different cancer types were from CPTAC. We adjusted the segments to a 100 kb window size, and the arm-level copy number alterations were calculated based on copy number package (*Nilsen et al., 2012*). We considered a log2-transformed copy number ratio > 0.2 as a gain and <(–0.2) as a loss. The aneuploidy degree corresponds to the total number of chromosome arm gains or losses (of any chromosome).

$$Aneuploidy\ score = count\ of\ gained\ or\ lost\ arms$$

The aneuploidy degree of CCLE was downloaded from geneDep website (*Cohen-Sharir et al., 2021*).

## Analysis of single-cell RNAseq dataset and quantification of gene expression variability

We analyzed only epithelial cells from Korean CRC patients (*Lee et al., 2020*), selecting the patients with a significant number of these cells (patient IDs: SMC16, SMC03, SMC09, SMC18, SMC21, SMC22). In order to quantify local gene expression variability, we applied the VarID method (*Grün, 2020*) from the RaceID3 package (v0.2.3) with default parameters, unless indicated. In brief, VarID defines locally homogeneous neighborhoods in cell state space, here set to 50 nearest-neighbors. UMI counts display a systematic variance–mean dependence, involving both biological and technical sources of variability, which is assumed to affect all genes similarly. VarID regresses out the variance–mean dependence by fitting a second-order polynomial to the baseline of the trend and subtracting it from the gene expression variance calculated for each locally homogenous neighborhood. The resulting corrected variance estimates allow to compare gene expression variability across different neighborhoods and across different genes, independently of their expression levels.

## Evaluation of the association of gene expression with aneuploidy by linear model

To find the genes whose RNA or protein expression changes along with aneuploidy, a linear model was used to fit the RNA or protein expression (for cell lines or individual tumor tissues) or the RNA or protein log2FC (for pooled tumor tissues) by aneuploidy score and other covariates including cancer types, copy number variation, purity, or cell cycle score. One example is shown as the following formula:

$$protein\ expression\ or\ log2FC \sim \beta_0 + \beta_1 \times aneuploidy\ score + \beta_2 \times purity$$

The t-value of aneuploidy coefficient $\beta_1$ was used to represent the association between RNA/protein level and aneuploidy degree with the control of other variables (such as purity). The genes were ranked based on the t-value of aneuploidy coefficient $\beta_1$, and then the enrichment of gene sets was calculated by GSEA with preranked module. C5 BP gene sets derived from the GO Biological Process ontology were used in those analyses. The gene sets whose size were smaller than 5 or bigger than 500 were removed before analyses.

As gene sets related to transcription and translation include many mitochondrial ribosome, rRNA, and tRNA genes, we also removed all mitochondrial genes before GSEA to exclude the possibility that mitochondrial genes overwhelmed those gene sets. For purity scores, CPTAC has two sets of purity data from nuclei percentage and the estimated amount of immune infiltrate based on the algorithm Estimate. Data from the algorithm Estimate are missing for ccRCC and OV. For the data from nuclei percentage, COAD, BRCA, and OV are missing. Those missing cancers were excluded from the pan-cancer analysis when purity was included in the model. The cell cycle score was calculated based on the average RNA level of 10 genes related to cell cycle entry (*Davoli et al., 2017*). To compare CPTAC and CCLE, the common genes of those two datasets were used for linear model and GSEA. The genes used to analyze changes at the RNA levels were the same ones used for analysis of protein change.

## Acknowledgements

We thank all the members of the Davoli and Fenyö labs as well as members of the Kelly Ruggles lab and Christine Vogel (NYU), Beatrix Ueberheide and Evgeny Kanshin (NYU Langone's Proteomics Laboratory) for helpful comments and insights during the completion of the project. We thank NYU Langone's Genome Technology Center and Proteomics Laboratory for help with RNAseq and the mass spectrometric experiments. *Figure 1A*, *Figure 2A*, *Figure 4A*, *Figure 5*, and *Figure 1—figure supplement 2B* were created with Biorender.com. This research was supported by a grant from the Cancer Research UK Grand Challenge, the Mark Foundation for Cancer Research (C5470/A27144), R00 CA212621 and R37 CA248631 to TD, the National Cancer Institute (NCI) Clinical Proteomic Tumor Analysis Consortium (CPTAC) grant U24CA210972 to DF, the NIH Institutional training grant T32GM136542, Training Program in Cell Biology to LK, the Cancer Center Support Grant P30CA016087 at the Laura and Isaac Perlmutter Cancer Center to NYU Langone's Genome Technology Center (RRID:SCR_017929) and Proteomics Laboratory (RRID:SCR_017926), and the German Research Foundation (322977937/GRK2344 MeInBio) and the ERC (818846 – ImmuNiche – ERC-2018-COG) to DG.

## Additional information

### Competing interests

Scott M Lippman: The other authors declare that no competing interests exist.

### Funding

| Funder | Grant reference number | Author |
| --- | --- | --- |
| Cancer Research UK | C5470/A27144 | Teresa Davoli |
| Mark Foundation For Cancer Research | C5470/A27144 | Teresa Davoli |
| National Cancer Institute | CA212621 | Teresa Davoli |
| National Cancer Institute | CA248631 | Teresa Davoli |
| National Cancer Institute | U24CA210972 | David Fenyo |
| National Institutes of Health | T32GM136542 | Lizabeth Katsnelson |
| National Cancer Institute | P30CA016087 | Teresa Davoli |
| German Research Foundation | 322977937/GRK2344 MeInBio | Dominic Grun |
| European Research Council | 818846 - ImmuNiche - ERC-2018-COG | Dominic Grun |

The funders had no role in study design, data collection and interpretation, or the decision to submit the work for publication.

### Author contributions

Pan Cheng, Conceptualization, Resources, Data curation, Software, Formal analysis, Validation, Investigation, Visualization, Methodology, Writing – original draft, Writing – review and editing; Xin Zhao, Conceptualization, Data curation, Software, Formal analysis, Validation, Visualization, Methodology, Writing – original draft, Writing – review and editing; Lizabeth Katsnelson, Validation, Visualization, Writing – original draft, Writing – review and editing; Elaine M Camacho-Hernandez, Validation, Writing – review and editing; Angela Mermerian, Validation; Joseph C Mays, Scott M Lippman, Writing – review and editing; Reyna Edith Rosales-Alvarez, Investigation, Methodology; Raquel Moya, Data curation, Methodology; Jasmine Shwetar, Methodology; Dominic Grun, David Fenyo, Conceptualization, Funding acquisition, Methodology, Writing – review and editing; Teresa Davoli,

Conceptualization, Resources, Supervision, Funding acquisition, Visualization, Methodology, Writing – original draft, Writing – review and editing

### Author ORCIDs
Pan Cheng http://orcid.org/0000-0002-9093-5066
Xin Zhao http://orcid.org/0000-0001-8502-0139
David Fenyo http://orcid.org/0000-0001-5049-3825
Teresa Davoli http://orcid.org/0000-0003-4116-9745

### Decision letter and Author response
Decision letter https://doi.org/10.7554/eLife.75227.sa1
Author response https://doi.org/10.7554/eLife.75227.sa2

---

# Additional files

### Supplementary files
• Supplementary file 1. Gene compensation analysis for *Figure 1*, *Figure 1—figure supplement 1*, and *Figure 1—figure supplement 2*.

• Supplementary file 2. DNA–RNA correlation and RNA–protein correlation analysis for *Figure 2* and *Figure 2—figure supplement 1*.

• Supplementary file 3. Complete list of cellular pathways and related analyses for *Figure 3*, *Figure 3—figure supplement 1*, and *Figure 3—figure supplement 2*.

• Supplementary file 4. Complete list of t-value and Gene Set Enrichment Analysis (GSEA) results for *Figure 4*.

• Transparent reporting form

### Data availability
The current manuscript is mainly a computational study using published datasets. Codes used in this manuscript are available in GitHub, https://github.com/davolilab/Proteogenomic-Analysis-of-Aneuploidy, (copy archived at swh:1:rev:9aa99245ac462b4134976293e52f56650ecb5c00). All other study data are included in the article and Supplementary files. For additional information and follow-up studies please also visit https://www.davolilab.com/.

The following previously published datasets were used:

| Author(s) | Year | Dataset title | Dataset URL | Database and Identifier |
|---|---|---|---|---|
| The Cancer Genome Atlas | 2005 | TCGA | https://portal.gdc.cancer.gov | TCGA, portal.gdc |
| Clinical Proteomics Tumor Analysis Consortium | 2017 | CPTAC2 | https://portal.gdc.cancer.gov/projects/CPTAC-2 | GDC Data Portal, CPTAC-2 |
| Clinical Proteomics Tumor Analysis Consortium | 2020 | CPTAC3 | https://portal.gdc.cancer.gov/projects/CPTAC-3 | GDC Data Portal, CPTAC-3 |
| The Genotype-Tissue Expression (GTEx) project | 2013 | GTEx | https://gtexportal.org/home/ | GTEx, gtexportal |
| the Cancer Cell Line Encyclopedia project | 2008 | CCLE | https://sites.broadinstitute.org/ccle/ | CCLE, broadinstitute |
| Genomic and Pharmacology Facilit, DTB, CCR, NCI, NIH | 2012 | NCI-60 | https://dtp.cancer.gov/discovery_development/nci-60/ | NCI-60, dtp.cancer |
| Wang D | 2019 | A deep proteome and transcriptome abundance atlas of 29 healthy human tissues | https://www.ebi.ac.uk/arrayexpress/experiments/E-MTAB-2836/ | ArrayExpress, E-MTAB-2836 |

| Author(s) | Year | Dataset title | Dataset URL | Database and Identifier |
|-----------|------|---------------|-------------|------------------------|
| Park WY | 2020 | Single cell RNA sequencing of colorectal cancer | https://ega-archive.org/datasets/EGAD00001005198 | European Genome-Phenome Archive, EGAD00001005198 |

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
