## [Editor Report]

The manuscript is of broad interest to researchers in the field of gene expression regulation and especially gene expression regulation in cancer cells. Gene expression can be regulated at several levels – in particular, the RNA and protein level. How each regulatory layer contributes to the final gene expression level is a central question in molecular biology. The authors tackle this fundamental question by asking how copy number variations at the level of DNA impact the other expression layers of RNA and protein. They do so mainly in a huge cohort of cancer samples, but also show that their findings extend to untransformed cells, and they find that there is rarely compensatory regulation at the RNA and protein level together, but that depending on the gene, expression is either compensated at the RNA level or protein level. This is an extensive meta-analysis of a huge set of samples that will be of interest to a broad readership.

---

## [Decision Letter]

**Decision letter after peer review:**

Thank you for submitting your article "Proteogenomic analysis of aneuploidy reveals divergent types of gene expression regulation across cellular pathways" for consideration by *eLife*. Your article has been reviewed by 2 peer reviewers, and the evaluation has been overseen by a Reviewing Editor and Naama Barkai as the Senior Editor. The following individual involved in the review of your submission has agreed to reveal their identity: Matthias Selbach (Reviewer #2).

Essential revisions:

1) The authors show that SCNAs are often significantly compensated at the protein level in most tumor types. This compensation is also normally stronger than RNA level compensation. A technical issue about this finding that needs to be addressed is that this is mainly based on proteomics data that used TMT for quantification. TMT-based quantifications, although quite precise, are not always the most accurate measurements in the sense of capturing the true amplitude of changes. This is due to the so-called ratio compression of TMT mass spec data. The authors need to account for that in order to exclude that this technical limitation of TMT-based proteomics measurements is a main contributor to the protein level compensation seen. Do the authors also have some proteomics data where label-free quantification of SILAC quantification was used? Do the same conclusions hold true when such data sets are used?

2) Many of the statistically significant differences seen – e.g complexed proteins versus non-complexed proteins, highly conserved proteins versus less conserved proteins – have actually a relatively small effect size. Rather than a bootstrapping strategy, it would be useful to also evaluate the differences using a Mann-Whitney U test.

*Reviewer #1 (Recommendations for the authors):*

– Figure 3A legend: for group 2 it should say "High RNA-protein correlation" instead of "Low RNA-protein correlation", shouldn't it?

– In Methods section lines 681 to 699. The data sets used should be described in more detail and not just by giving direct links to them. E.g. what is the quantification method for proteomics data used, etc.? This is important to evaluate the analysis for potential technical artifacts due to data collection in the different data sets.

– In the "Methods" section at line 732 – "random sampling the CS" – how big was the sample each time? This is not just here but throughout the analysis part where bootstrapping is used.

– In the "Methods" section lines 765 to 772 – to be honest I do not fully understand what the authors did here. Could you maybe rephrase this section?

– In the "Methods" section line 891 – the peptides were TMT labeled. Therefore, I do not think DIA measurements were done but rather DDA – should that maybe mean "(DDA)" instead of "DIA"?

– In the "Methods" section line 915 – it indicates that in MaxQuant the "Match between the runs" feature was on. What is the benefit of that if TMT samples were measured as an MS2 spectrum anyway needs to be recorded to get quantitative information? Did the authors use another program in addition, like Dart-ID?

*Reviewer #2 (Recommendations for the authors):*

1. Ribosomal proteins make up a significant fraction of proteins that are overproduced and show protein-level compensation in aneuploid cells. Did the authors check how (i) ribosomal proteins look like as a group and (ii) how the data changes if ribosomal proteins are excluded from the analyses? This is to assess whether the findings are dominated by this specific subset of proteins.

2. One technical limitation of the TMT multiplexes proteomic data is ratio compression. Due to this effect, the observed absolute log2FC tends to be smaller than true log2FCs. This technical artifact might be mist-interpreted as protein-level compensation. Please mention and discuss this potential limitation.

3. Line 123: "Dosage compensation is a process by which cells modulate gene expression to buffer against changes in DNA copy number" – I think dosage compensation is defined in the context of sex chromosomes – a mechanism to ensure that the homogametic sex does not have too much or the heterogametic sex too little of the gene products. I do not think the term should be used in the context of aneuploidy.

4. Line 138: "For each gene of each cancer type, we defined the samples that did not have DNA copy number changes (log2 copy number ratio between -0.2 to 0.2) as the neutral group." How are these DNA copy number changes normalized? How did the authors deal with possible whole genome doubling in cancer? This question is relevant because it affects the size of relative changes: For example, going from 2 copies (diploid cancer) to 3 copies (for amplified regions) is a larger relative gain than from 4 copies (cancer with whole genome doubling) to 5.

5. Line 554: "The protein compensation for complex genes of DNA gains is thought to occur through protein degradation of the overabundant subunits (McShane et al., 2016). However, this model cannot easily explain how protein compensation happens after DNA losses and why the compensation is stronger for protein complex genes." I disagree with this point: The model can (to some extent) also explain compensation after DNA loss. The key point is that overproduction of proteins does not only occur during aneuploidy but is a widespread feature even in euploid cells: Many subunits of multiprotein complexes are overproduced (and rapidly degraded) in diploid cells. This baseline overproduction buffers proteins against gene copy number losses: Loss of one copy for such will result in reduced protein overproduction (and reduced degradation). But as long as the overproduction (at baseline) is greater than the reduction due to the DNA-level loss there should be full compensation. One way to assess this would be to look at how the protein compensation upon DNA loss correlates with the degree of protein overproduction in diploid cells. Specifically, the fraction of protein overproduction (and rapid degradation) in diploid RPE-1 cells can be easily computed from the Markov-chain based model for non-exponential protein degradation (see Figure 2 plus legend in Taggart et al., 2020 for the formula and Table S4 from McShane et al., 2016 for model parameters). Assuming this overproduction is to some extent similar in different cells, I would expect that protein compensation upon DNA loss correlates with "baseline" protein overproduction in diploid cells.

6. Line 586 and following: This is the Discussion section, and the authors are of course free to speculate about the biological meaning of their findings. Having said this, I have different opinions on a number of points they may want to consider. First, I do not think that energy conservation can explain RNA-level regulation in a satisfying way: The energy cost to synthesise and degrade mRNAs is negligible relative to the cost to synthesise and degrade proteins (see for example figure S12C in Schwanhausser et al., Nature, 2011). Second, I do not think that the faster speed of regulation can explain mRNA level regulation: In contrast to the statement made in the discussion, regulation at the protein level (translation or protein degradation) enables faster changes in protein levels than changes at the mRNA level (see DOI: 10.1002/bies.201300017, for example). In contrast to these explanations, I think it is helpful to see protein-level regulation as a consequence of the missing mRNA-level regulation: Some genes may be gene-specific regulatory feedback mechanisms regulating mRNA levels. These genes do not have much protein-level control because copy number changes are already buffered at the mRNA level. For example, as nicely pointed out by the authors, protein-level control is difficult for secreted proteins, which means that there is evolutionary pressure to evolve mRNA-level feedback mechanisms. In contrast, genes w/o such mRNA level buffering are buffered at the protein level. The degradation of orphan protein complex subunits provides a mechanistic explanation of how this could be achieved. I think it is also helpful to think about how regulation can mechanistically occur, given that there is no known universal mechanism that "measures" mRNA or protein levels and adjusts transcription and translation accordingly. In my opinion, RNA-level regulation evolved because (i) this regulation is functionally important (like for genes encoding secreted proteins) and (ii) because regulatory feedback is mechanistically feasible (like transcription factors regulating their own transcription, RNA-binding proteins regulating stability of their own RNA). Other genes which did not have gene-specific regulatory feedback loops remain unbuffered or are buffered at the protein level (where the degradation of orphan subunits via ligases like UBE2O provides a universal mechanism for protein-level buffering). Some of these points are also discussed in a recent review (see below).

7. The authors may want to add these two relevant recent papers – Senger G, Schaefer MH. 2021. Protein Complex Organization Imposes Constraints on Proteome Dysregulation in Cancer. Frontiers in Bioinformatics. 1:33- Buccitelli C, Selbach M. 2020. mRNAs, proteins and the emerging principles of gene expression control. Nat Rev Genet. 630-644.

---

## [Author Response]

Essential revisions:1) The authors show that SCNAs are often significantly compensated at the protein level in most tumor types. This compensation is also normally stronger than RNA level compensation. A technical issue about this finding that needs to be addressed is that this is mainly based on proteomics data that used TMT for quantification. TMT-based quantifications, although quite precise, are not always the most accurate measurements in the sense of capturing the true amplitude of changes. This is due to the so-called ratio compression of TMT mass spec data. The authors need to account for that in order to exclude that this technical limitation of TMT-based proteomics measurements is a main contributor to the protein level compensation seen. Do the authors also have some proteomics data where label-free quantification of SILAC quantification was used? Do the same conclusions hold true when such data sets are used?

We thank the reviewers (see similar comment below from the other reviewer) for this comment and point which we have now addressed through the following literature search or analyses:

First, we found there are some previous studies which observed the similar protein-level compensation in yeast and human cells by different detection methods. Dephoure et al. compared two different methods, stable isotope labeling by amino acids in cell culture (SILAC) and tandem mass tag (TMT) based proteomics. The protein-level compensation of gained genes in yeast was discovered by both methods (Figure 2 and Figure 2 —figure supplement 1 of Dephoure et al., 2014). Similarly, Stingele et al. identified the protein-level compensation in pairs of isogenic diploid and aneuploid human cell lines by SILAC (Figure 2B of Stingele et al., 2012). Another group also found the protein-level compensation in primary human fibroblasts from individuals with Patau (trisomy 13), Edwards (trisomy 18) or Down (trisomy 21) syndromes by MS3-based approach (Hwang et al., 2021), which should eliminate the interference of ratio distortion (Ting et al., 2011). Taken together, those previous studies suggest the protein-level compensation should not be just the artifacts induced by the technical limitation of TMT-based proteomics.

To further validate the protein-level compensation, we performed the same analysis on TCGA (The Cancer Genome Atlas Program) (Research Network et al., 2013) COAD samples for which label-free proteomics data is available (Zhang et al., Nature, 2014). Consistent with TMT-based proteomics, significant compensation at the protein level was found, which is higher for complex genes than non-complex genes (Figure 1 —figure supplement 1C, Supplementary File 1G). As we observed before for COAD (Figure 1C), RNA-level compensation was shown in all groups of DNA change, and was stronger for non-complex genes (deep loss and high gain, FDR<0.005, Figure 1 —figure supplement 1C, Supplementary File 1G). These results suggest that the limitations imposed by the TMT quantification do not alter the conclusions of our analysis on gene compensation. We have now added this data in Figure 1 —figure supplement 1C and Supplementary File 1G and corresponding text at page 5.

2) Many of the statistically significant differences seen – e.g complexed proteins versus non-complexed proteins, highly conserved proteins versus less conserved proteins – have actually a relatively small effect size. Rather than a bootstrapping strategy, it would be useful to also evaluate the differences using a Mann-Whitney U test.

We thank the reviewers for this comment, and we have addressed it in detail. We have performed the analyses using Mann-Whitney U test and Kolmogorov-Smirnov (KS) test (Supplementary File 2K). Compared with bootstrapping, the p-values calculated by Mann-Whitney U test or KS test were much smaller, close to zero. While Mann-Whitney U test or KS test carries the risk of p-value inflation due to the high sample number, the bootstrapping method can solve the problem as it is independent from the sample number. Initially we had used Mann-Whitney U test for all our analyses and were suggested to include bootstrapping method after consultation with the NYU Biostatistics Resource.

For this revised manuscript, we added a new result related to the impact of distinct pattern of gene regulation on single-cell gene expression. We asked whether the genes of distinct regulation, which we found based on the bulk RNA-seq and mass spectrometry data, also show different regulation at single-cell level. We assayed the level of variability in the RNA level across individual cells by using VarID (Grün, 2020), a computational method that quantifies gene expression variability locally in cell state space. We analyzed single-cell RNAseq data from 6 patients with colorectal cancer (CRC) (Lee et al., 2020). Our analysis shows that Group 2 genes (low DNA-RNA correlation and high RNA-protein correlation), preferentially regulated at the RNA level, tend to have higher expression variability than the Group 1 genes (high DNA-RNA correlation and low RNA-protein correlation) which are predominantly regulated on the protein level (Figure 3F). We have now added this data in Figure 3F and corresponding text at page 11.

Finally, we have We have added a figure, Figure 5 with the goal of conveying the main message of the paper in a more effective way.

Reviewer #1 (Recommendations for the authors):– Figure 3A legend: for group 2 it should say "High RNA-protein correlation" instead of "Low RNA-protein correlation", shouldn't it?

We agree and we have changed the text accordingly.

– In Methods section lines 681 to 699. The data sets used should be described in more detail and not just by giving direct links to them. E.g. what is the quantification method for proteomics data used, etc.? This is important to evaluate the analysis for potential technical artifacts due to data collection in the different data sets.

We thank the reviewer for this comment, and we have now added a much more detail description of the data sets used (see also Methods).

For BRCA and LUAD the Spectrum Mill software package v7.0 pre-release (Agilent Technologies, Santa Clara, CA) was used for MS data analysis. Protein identification was performed by searching the MS/MS spectra against protein sequence database obtained using the UCSC Table Browser (https://genome.ucsc.edu/cgi-bin/hgTables) on September 14, 2016, that contains 37,579 proteins mapped to the human reference genome (hg19), adding common contaminants, mitochondrial proteins, and non-canonical small open reading frames. The searches were performed allowing ± 20 ppm mass tolerance for precursor and product ions, allowing for common modification. Peptide spectrum matches (PSMs) were filtered for 30% minimum matched peak intensity and target-decoy-based false discovery rate (FDR) estimates at the PSM level, and for proteins protein level for each TMT-plex for all TMT-plexes for a tumor type, and for phospho all TMT-plexes for a tumor type, and for phosphorylation at the site levels. Normalization of each peptide was performed using the common reference, and a 2-component Gaussian mixture model-based normalization was used to nullify the effect of differential protein loading and/or systematic MS variation.

For COAD, OV, UCEC MS-GF+ v9881 (Gibbons et al., 2015, Kim and Pevzner, 2014, Kim et al., 2008) was used to search against the RefSeq human protein sequence database downloaded on June 29, 2018 (hg38; 41,734 proteins), combined with 264 contaminants (e.g., trypsin, keratin) using partial tryptic peptides, ± 10 ppm parent and fragment ion tolerance, allowing for isotopic error in precursor ion selection and common modifications (static carbamidomethylation (+57.0215 Da) on Cys residues and TMT modification (+229.1629 Da) on the peptide N terminus and Lys residues, and dynamic oxidation (+15.9949 Da) on Met residues), and including decoy sequences generated by reversing the protein sequences. Peptides were filtered using a maximum false discovery rate (FDR) of 1% at peptide level using PepQValue < 0.005 and parent ion mass deviation < 7 ppm criteria. A minimum of 6 unique peptides per 1000 amino acids of protein length was required for achieving 1% at the protein level within the full dataset. The TMT reporter ion intensities were extracted using MASIC (Monroe et al., 2008). Relative protein levels were calculated as the ratio of sample abundance to reference abundance using the summed reporter ion intensities from peptides that could be uniquely mapped to a gene. The relative abundances were log2 transformed and zero-centered for each gene to obtain final relative abundance values. Each sample was median centered to adjust for differences in laboratory conditions and sample handling.

For HNSC, PAD and ccRCC the MSFragger version 3.0 (Kong et al., 2017) was used to search the RefSeq human protein sequences and an equal number of decoy sequences using tryptic and semi-tryptic peptides allowing two missed cleavages, a mass tolerance of 10 ppm, and allowing isotope errors, mass calibration, spectral deisotoping, and parameter optimization (Yu et al., 2020). Cysteine carbamidomethylation, lysine and peptide N-terminal TMT labeling were specified as fixed modifications, and Methionine oxidation and serine TMT labeling were specified as variable modifications, and for the phosphopeptide enriched data, phosphorylation of serine, threonine, and tyrosine residues was allowed. Philosopher toolkit version v3.2.8 (da Veiga Leprevost et al., 2020) was used for post-processing. The protein groups assembled by ProteinProphet (Nesvizhskii et al., 2003) were filtered to 1% protein-level False Discovery Rate (FDR). To generate summary reports TMT-Integrator (Djomehri et al., 2020) was used. PSMs mapping to common contaminant proteins was excluded, and both unique and razor peptides were used for quantification. The reporter ion intensities of each PSM were log2 transformed and normalized by the reference channel intensity median centered after removal of outliers.

– In the "Methods" section at line 732 – "random sampling the CS" – how big was the sample each time? This is not just here but throughout the analysis part where bootstrapping is used.

In each bootstrapping test, we chose the sample sizes that are the same as the original groups and repeated the sampling for 10,000 times. We have now added more details in the corresponding parts in the methods. And a specific section called bootstrapping strategy has been added in the methods.

– In the "Methods" section lines 765 to 772 – to be honest I do not fully understand what the authors did here. Could you maybe rephrase this section?

We apologize for having a description without enough details. We have now extended the description significantly as the following (Methods, page 20).

To estimate the association between the DNA-RNA (DR) and RNA-protein (RP) correlations at the gene level (Figure 3A), we first calculated the pan-cancer DR and RP Spearman’s correlations (rho values) for each gene resulting in a density distribution f(DR, RP). We split the DR range into a series of windows (40 bins), and in each of the windows, i, the RP value of the maximum density, RPi, was chosen to represent the RNA-protein correlation of genes in the windows, i.e.RPi=argmaxRPf(DRi,RP)

Therefore, a series of representative points were determined: (DRi,RPi), i=1, 2, …, 40. The slope of those representative points was used as an estimate of the association between DNA-RNA and RNA-protein correlations.

– In the "Methods" section line 891 – the peptides were TMT labeled. Therefore, I do not think DIA measurements were done but rather DDA – should that maybe mean "(DDA)" instead of "DIA"?

We apologize for this typo (which we have now fixed) and thank the reviewer for this comment.

– In the "Methods" section line 915 – it indicates that in MaxQuant the "Match between the runs" feature was on. What is the benefit of that if TMT samples were measured as an MS2 spectrum anyway needs to be recorded to get quantitative information? Did the authors use another program in addition, like Dart-ID?

The "Match Between Runs" (MBR) option started to make sense for TMT data since the MaxQuant release 1.16.12.0. The improved algorithm is described in Sung-Huan Yu et al., 2020 and it allows to extract TMT quantification data for peptides that were sequenced by MS/MS but not identified due to low spectra quality (but identified with a good MS/MS in another run and matched through MBR).

Reviewer #2 (Recommendations for the authors):1. Ribosomal proteins make up a significant fraction of proteins that are overproduced and show protein-level compensation in aneuploid cells. Did the authors check how (i) ribosomal proteins look like as a group and (ii) how the data changes if ribosomal proteins are excluded from the analyses? This is to assess whether the findings are dominated by this specific subset of proteins.

We thank the reviewer for this comment. Indeed, ribosomal proteins make up a substantial fraction of proteins in protein complexes and whether our results are dependent on their high representation among complexes is a good point. We have addressed this question in 3 ways and generally found that excluding the ribosomal genes from the protein complex genes does not alter the results of our analyses.

For Figure 1 we have repeated the pan-cancer analysis separating ribosomal complex genes and other complex genes as reported in Figure 1 —figure supplement 1B and described the results at page 5 (which are consistent with our original observation). A brief summary is provided here:

We have repeated the analysis related to Figure 1B. Both ribosomal and non-ribosomal complex genes showed significant compensation at the protein level for both gains and losses. Strikingly, the protein-level compensation of ribosomal genes was so strong that the median protein log2FC remained almost unchanged for high gains and deep losses; this was not the case for the RNA level (Figure 1 —figure supplement 1B, Supplementary File 1F). Such kind of compensation was not observed at the RNA level except for the group of high DNA gain. For the high DNA gain group, non-ribosomal complex genes have lower RNA-level compensation than non-complex genes, consistent with our previous observations. We have added these data to Figure 1 —figure supplement 1B and Supplementary File 1F.

For Figure 2 we have repeated the analysis excluding ribosomal genes as reported in Figure 2 —figure supplement 1C and described the data which are consistent with our original observation at page 7.

For Figure 3B, one of the most important figures/findings, the original analyses were already done *by pathway* which should not pose a problem for the point raised here.

2. One technical limitation of the TMT multiplexes proteomic data is ratio compression. Due to this effect, the observed absolute log2FC tends to be smaller than true log2FCs. This technical artifact might be mist-interpreted as protein-level compensation. Please mention and discuss this potential limitation.

The other reviewer also raised this very point and we thank the reviewer for this comment. We have now addressed it through the following literature search or analyses:

First, we found there are some previous studies which observed the similar protein-level compensation in yeast and human cells by different detection methods. Dephoure et al. compared two different methods, stable isotope labeling by amino acids in cell culture (SILAC) and tandem mass tag (TMT) based proteomics. The protein-level compensation of gained genes in yeast was discovered by both methods (Figure 2 and Figure 2 —figure supplement 1 of Dephoure et al., 2014). Similarly, Stingele et al. identified the protein-level compensation in pairs of isogenic diploid and aneuploid human cell lines by SILAC (Figure 2B of Stingele et al., 2012). Another group also found the protein-level compensation in primary human fibroblasts from individuals with Patau (trisomy 13), Edwards (trisomy 18) or Down (trisomy 21) syndromes by MS3-based approach (Hwang et al., 2021), which should eliminate the interference of ratio distortion (Ting et al., 2011). Taken together, those previous studies suggest the protein-level compensation should not be just the artifacts induced by the technical limitation of TMT-based proteomics.

To further validate the protein-level compensation, we performed the same analysis on TCGA (The Cancer Genome Atlas Program) (Research Network et al., 2013) COAD samples for which label-free proteomics data is available (Zhang et al., Nature, 2014). Consistent with TMT-based proteomics, significant compensation at the protein level was found, which is higher for complex genes than non-complex genes (Figure 1 —figure supplement 1C, Supplementary File 1G). As we observed before for COAD (Figure 1C), RNA-level compensation was shown in all groups of DNA change, and was stronger for non-complex genes (deep loss and high gain, FDR<0.005, Figure 1 —figure supplement 1C, Supplementary File 1G). These results suggest that the limitations imposed by the TMT quantification do not alter the conclusions of our analysis on gene compensation. We have now added this data in Figure 1 —figure supplement 1C and Supplementary File 1G and corresponding text at page 5.

3. Line 123: "Dosage compensation is a process by which cells modulate gene expression to buffer against changes in DNA copy number" – I think dosage compensation is defined in the context of sex chromosomes – a mechanism to ensure that the homogametic sex does not have too much or the heterogametic sex too little of the gene products. I do not think the term should be used in the context of aneuploidy.

We thank the reviewer for this comment. We agree that “dosage compensation” is defined in the context of sex chromosomes even though sometimes it is used for autosomal chromosomes as well (Hose et al., 2015, Brennan et al., 2019, Siegel and Amon et al., 2012). To avoid misunderstandings, we used gene or protein compensation rather than dosage compensation in the manuscript.

4. Line 138: "For each gene of each cancer type, we defined the samples that did not have DNA copy number changes (log2 copy number ratio between -0.2 to 0.2) as the neutral group." How are these DNA copy number changes normalized? How did the authors deal with possible whole genome doubling in cancer? This question is relevant because it affects the size of relative changes: For example, going from 2 copies (diploid cancer) to 3 copies (for amplified regions) is a larger relative gain than from 4 copies (cancer with whole genome doubling) to 5.

We thank the reviewer for this comment. In general, the copy number refers to the Log2 copy number ratio – defined as the log2 of the ratio between the copy number of the gene and the average copy number of the rest of the genome and is independent from the ploidy. So it is normalized to the average genome copy number and it reflects the fractional change in copy number: for example, it is the same for a diploid cells losing one copy and for a tetraploid cells losing 2 copies.

We agreed that genome doubling may be a problem when we calculate the size of relative changes. However, we couldn’t distinguish samples of genome doubling in CPTAC database because of the lack of such information. To exclude the interference of genome doubling, we analyzed the proteomics data of TCGA samples (Zhang et al., Nature, 2014, Mertins et al., Nature, 2016, Zhang et al., Cell, 2016) and the conclusions hold true after the samples of genome doubling were removed from the analysis, as reported in the text (Figure 1 —figure supplement 1D. and page 6). Therefore, these data indicate that the presence of genome doubling in a fraction of the samples does not affect the results of our analyses.

5. Line 554: "The protein compensation for complex genes of DNA gains is thought to occur through protein degradation of the overabundant subunits (McShane et al., 2016). However, this model cannot easily explain how protein compensation happens after DNA losses and why the compensation is stronger for protein complex genes." I disagree with this point: The model can (to some extent) also explain compensation after DNA loss. The key point is that overproduction of proteins does not only occur during aneuploidy but is a widespread feature even in euploid cells: Many subunits of multiprotein complexes are overproduced (and rapidly degraded) in diploid cells. This baseline overproduction buffers proteins against gene copy number losses: Loss of one copy for such will result in reduced protein overproduction (and reduced degradation). But as long as the overproduction (at baseline) is greater than the reduction due to the DNA-level loss there should be full compensation. One way to assess this would be to look at how the protein compensation upon DNA loss correlates with the degree of protein overproduction in diploid cells. Specifically, the fraction of protein overproduction (and rapid degradation) in diploid RPE-1 cells can be easily computed from the Markov-chain based model for non-exponential protein degradation (see Figure 2 plus legend in Taggart et al., 2020 for the formula and Table S4 from McShane et al., 2016 for model parameters). Assuming this overproduction is to some extent similar in different cells, I would expect that protein compensation upon DNA loss correlates with "baseline" protein overproduction in diploid cells.

We thank the reviewer for this very interesting point and idea. We fully agree with the reviewer that overproduction happens also in normal cells and needs to be regulated; this is one of the reasons why we think that the type of regulation (RNA vs protein level) defined in aneuploid cells may reflect general rules of regulation. We also agree that our statement regarding regulation of protein level after DNA loss is rather speculative and not supported by data. The idea proposed by the reviewer on a mechanism of protein compensation for losses dependent on protein overproduction is a very appealing one and we have now stated this possibility in the Discussion. We also tested the idea following the reviewer’s advice in three different datasets: HNSC (head and neck cancer) data and UCEC (uterine cancer) data, chosen because they have a strong protein-level but a weak RNA-level compensation for gene of DNA losses (Figure 1C) and also a proteogenomic data from untransformed cells (RPE) containing chromosome losses (Chunduri et al., 2021). In each case, we considered the complex genes that have DNA losses, and we have studied the correlation between the compensation score (CS) and the fraction of protein overproduction calculated based on the formula from Taggart et al., 2020 and model parameters from the McShane 2016 paper. In neither of these datasets, we were able to find a strongly positive correlation between the two parameters as shown in Author response image 1. This may be due to intrinsic limitations of the datasets (such as number of genes on the monosomic chromosome) that may prevent to see an association; but we have decided not to include this analysis in the manuscript given the difficulty in fully interpreting it. Nevertheless, we have added this idea/comment to the Discussion.

**Author response image 1. sa2fig1:** 

6. Line 586 and following: This is the Discussion section, and the authors are of course free to speculate about the biological meaning of their findings. Having said this, I have different opinions on a number of points they may want to consider. First, I do not think that energy conservation can explain RNA-level regulation in a satisfying way: The energy cost to synthesise and degrade mRNAs is negligible relative to the cost to synthesise and degrade proteins (see for example figure S12C in Schwanhausser et al., Nature, 2011). Second, I do not think that the faster speed of regulation can explain mRNA level regulation: In contrast to the statement made in the discussion, regulation at the protein level (translation or protein degradation) enables faster changes in protein levels than changes at the mRNA level (see DOI: 10.1002/bies.201300017, for example). In contrast to these explanations, I think it is helpful to see protein-level regulation as a consequence of the missing mRNA-level regulation: Some genes may be gene-specific regulatory feedback mechanisms regulating mRNA levels. These genes do not have much protein-level control because copy number changes are already buffered at the mRNA level. For example, as nicely pointed out by the authors, protein-level control is difficult for secreted proteins, which means that there is evolutionary pressure to evolve mRNA-level feedback mechanisms. In contrast, genes w/o such mRNA level buffering are buffered at the protein level. The degradation of orphan protein complex subunits provides a mechanistic explanation of how this could be achieved. I think it is also helpful to think about how regulation can mechanistically occur, given that there is no known universal mechanism that "measures" mRNA or protein levels and adjusts transcription and translation accordingly. In my opinion, RNA-level regulation evolved because (i) this regulation is functionally important (like for genes encoding secreted proteins) and (ii) because regulatory feedback is mechanistically feasible (like transcription factors regulating their own transcription, RNA-binding proteins regulating stability of their own RNA). Other genes which did not have gene-specific regulatory feedback loops remain unbuffered or are buffered at the protein level (where the degradation of orphan subunits via ligases like UBE2O provides a universal mechanism for protein-level buffering). Some of these points are also discussed in a recent review (see below).

We thank the reviewer for these very insightful comments. Based on these comments, we have edited the Discussion as follows.

We have added a figure, Figure 5 with the goal of conveying the main message in a more effective way.

Regarding the point of energy conservation, we have realized that our statement was not clear enough. Although energy demand is much less for RNA synthesis than for protein synthesis, our idea was referring to the fact that to achieve RNA-level regulation (let’s say transcriptional regulation) of the subunits of a protein complex, a complex mechanism/system is likely required. The maintenance and function of such a complex system may be more energy-consuming. In fact, for genes protein complexes, by “high degree of RNA-level regulation” we mean that the level of mRNA of all complex subunits is adjusted/buffered with respect to their DNA copy number alteration: if let’s say subunit A is gained compared to subunit B, the transcriptional output of gene A would need to be adjusted to the one of B (in the hypothesis of high regulation). This would be particularly difficult for a complex of many subunits as there would need to be a flow of information from the gene copy number of one subunit to the others. This is consistent with the reviewer’s idea that cells need gene-specific regulatory feedback mechanisms regulating mRNA levels which would be very complicated and energy-consuming if mRNA of every subunits need to be adjusted relative to the DNA copy number of the others. Therefore, this ‘RNA-regulation’ would require much more energy than simply RNA synthesis, in that it would require a complex regulatory system. On the other hand, degradation of unstable subunits may be achieved through a simpler mechanism, as in the case of UBE2O mentioned by the reviewer. This is what we referred to when we mentioned the “energy constraint”, referring not only to the energy demand of transcription versus translation/degradation but to the overall energy required to put in place a gene regulatory network to allow RNA-level regulation of subunits of multiprotein complexes. In a way, this is analogous to the reviewer point on the fact that for certain genes RNA regulation is not feasible as mentioned in the reviewer point “RNA-level regulation evolved because …. and (ii) because regulatory feedback is *mechanistically feasible* …”. We have modified the Discussion to better reflect this point in a more clear way.

We agree with the reviewer on the statement that regulation at the protein level (translation or protein degradation) generally enables faster changes in protein levels than regulation at the mRNA level and we have removed this sentence.

Regarding the following point on that the fact that “it is helpful to see protein-level regulation as a consequence of the missing mRNA-level regulation”, we also agree and in a way this exactly the take-home message of the paper where we show that across pathways the level of RNA regulation is inversely proportional to the level of protein regulation. We have modified the Discussion to make this point even more clear.

7. The authors may want to add these two relevant recent papers – Senger G, Schaefer MH. 2021. Protein Complex Organization Imposes Constraints on Proteome Dysregulation in Cancer. Frontiers in Bioinformatics. 1:33- Buccitelli C, Selbach M. 2020. mRNAs, proteins and the emerging principles of gene expression control. Nat Rev Genet. 630-644.

We thank the reviewer and have now added these references.